

# A global climatology of sting-jet extratropical cyclones

Suzanne L. Gray[1], Ambrogio Volonté[1,2], Oscar Martínez-Alvarado[1,2], and Ben J. Harvey[1,2]

[1]University of Reading, Reading, UK
[2]National Centre for Atmospheric Science, University of Reading, Reading, UK

**Correspondence:** Suzanne L. Gray (s.l.gray@reading.ac.uk)

**Abstract.** Sting jets have been identified in the most damaging extratropical cyclones impacting northwest Europe. Unlike the cold conveyor belt and other long-lived cyclone wind jets, sting jets can lead to regions of exceptionally strong near-surface winds, and damaging gusts, over just a few hours and with much smaller wind "footprints". They descend into the frontal-fracture region found in warm-seclusion cyclones. Previous research has focused almost exclusively on North Atlantic-

European cyclones, but there are no known physical reasons why sting jets should not develop elsewhere and recognition of their existence can inform weather nowcasting and wind warnings. We have produced the first climatology of sting-jet cyclones over the major ocean basins. A sting-jet precursor diagnostic has been applied to more than 10,000 warm-seclusion cyclones in the top intensity decile, tracked using 43 extended-winters of ERA5 reanalysis data. Cyclones with sting-jet precursors are found to occur over the North Pacific and Southern Oceans for the first time and they are more prevalent in the Northern

Hemisphere (27% of all top decile cyclones) compared to the Southern Hemisphere (15%). These cyclones have distinct characteristics to those without the precursor including initiating closer to the equator, deepening faster in mean-sea-level pressure and having stronger near-surface winds, even in the reanalysis data which is too coarse to resolve sting jets. Composite analysis reveals systematic differences in structural evolution, including in potential vorticity and jet crossing. These differences evidence the climatological consequences of strong diabatic cloud processes on cyclone characteristics, implying that sting jets

are likely to be enhanced by climate change.

## 1 Introduction

Sting jets have been identified in several of the most damaging extratropical cyclones impacting northwest Europe (see the review paper by Clark and Gray (2018)) since they were first identified in 2004 from a re-examination of observations of

the 1987 Great October Storm that devastated southern England (Browning, 2004; Clark et al., 2005). As a recent example, Volonté et al. (2023a, b) present evidence of the sting jets in storm Eunice, which caused extensive wind damage in southern England in February 2022. The possibility of sting jets enhancing damaging winds and gusts is now routinely considered by the (UK) Met Office in forecasting and generating weather warnings, and the term "sting jet" is used in their communications and by the British media. The aim of this study is to identify and characterise sting-jet cyclones, and their distinctiveness to





non-sting-jet cyclones, in the three major ocean basins. Recognition of their potential existence is a first step towards improved
nowcasting and warning of associated wind risk. Sting jets are also a consideration for the currently uncertain projections of
climate change effects on cyclone wind intensity (Catto et al., 2019).

Strong near-surface winds in extratropical cyclones are mainly associated with one of two major airflows termed conveyor
belts: the warm conveyor belt ahead of the cold front, and the cold conveyor belt poleward of the cyclone centre that travels

rearwards relative to the cyclone motion. Their definitions, and those of other major cyclone features, date back to at least
the 1980s (see Browning and Roberts (1994)) and the associated surface winds typically last many hours, yielding windspeed
"footprints", up to 100s of km wide by 1000s of km long. In contrast, sting jets are, as defined by Clark and Gray (2018),
coherent airflows that descend over just a few hours from within the cloud heads visible in intense cyclones. They are distinct
from the conveyor belt jets, although they can interact with the cold conveyor belt jet, and can lead to regions of exceptionally

strong near-surface winds, and particularly damaging gusts, with much smaller footprints (50–100 km across; Hewson and
Neu (2015)). Their small-scale, transient nature is one reason for their much more recent identification: they can pass between
operational surface weather stations, although they have occasionally been captured by wind profilers (Volonté et al., 2018;
Parton et al., 2009) and Doppler radar (Browning et al., 2015), and their full representation by numerical models require
the latest high resolution weather forecast models. Sting jets descend into the frontal-fracture region of cyclones developing

according to the Shapiro-Keyser conceptual model (Shapiro and Keyser, 1990). One distinctive aspect of Shapiro-Keyser
cyclones is that a seclusion of warm air forms in their centre as they approach maturity, which is delimited by a bent-back
warm front wrapping around the poleward side of the low-pressure centre; this structure contrasts with the occluded front
characteristic of cyclones that develop according to the other major conceptual model: the Norwegian model.

Since their identification, knowledge of the characteristics and physical mechanisms causing sting jets has advanced con-

siderably. Analysis of cyclone case studies and idealised model simulations led to the development of a conceptual model
of sting-jet evolution, revealing the role of mesoscale instabilities, including conditional symmetric instability (CSI), in their
generation, descent and acceleration (Gray et al., 2011; Volonté et al., 2020, 2018). A climatological analysis using a precursor
diagnostic based on the potential for CSI release by descent in the cloud head showed that sting jets exist in perhaps one in
three North Atlantic cyclones, particularly explosive ones (Hart et al., 2017). Sting-jet frequency is projected to increase in the

future (Martinez-Alvarado et al., 2018; Manning et al., 2021, 2023). However, all this research has focused exclusively on cy-
clones crossing the North Atlantic and affecting northwest Europe. A single case of sting-jet activity in a cyclone originating in
the Mediterranean has recently been published (Brâncuş et al., 2019), but no published cases conclusively diagnosing sting-jet
existence exist for other extratropical regions and there is published climatology beyond the North Atlantic. Despite this, there
are no known physical reasons why sting jets should not develop over other extratropical oceanic basins.

In developing this climatology we address two sets of questions related to their locations and structures, respectively:

1.  Is the sting jet a global extratropical phenomenon or is it limited to North Atlantic cyclones? What are the tracks and
prevalence of these cyclones over the major ocean basins and how do they differ from those of non-sting-jet cyclones?





2.  What are the structural differences between sting-jet and non-sting-jet cyclones, globally and in regional subsets? For example, are sting-jet cyclones more intense as is found for cyclones crossing the North Atlantic (Hart et al., 2017)

Unambiguous identification of sting jets in cyclone case studies requires detailed analysis of numerical weather prediction model output from a model with high enough resolution to resolve the sting jet: Clark and Gray (2018) conclude a minimum horizontal grid spacing of 10–15 km is required with correspondingly high enough vertical grid spacing to resolve the slantwise motions. The analysis ideally includes analysis of air parcel trajectories to disentangle the low-level wind jets and observational analysis to support the model prediction. Thus, generation of a climatology requires an alternative approach, both due to

the challenge of identifying sting jets and because such analysis typically makes use of long-term reanalysis products, or climate model outputs, produced using models that have insufficient resolution to resolve sting jets. Rather than unambiguously identifying sting jets in cyclones, instead cyclones that are likely to contain sting jets can be identified.

Here a sting-jet precursor (SJP) diagnostic based on the presence of CSI within the cloud head, developed by Martínez-Alvarado et al. (2013) and previously used to produce the first North Atlantic climatologies of sting-jet cyclones (Martínez-

Alvarado et al., 2012; Hart et al., 2017), has been adapted and applied to tracked cyclones with warm seclusions in 43 years of extended-winter ERA5 reanalysis data across the entire extratropics in both hemispheres. In addition to production of the first global climatology of extratropical sting-jet cyclones, novel aspects of this study include the development of a new approach to identify cyclones containing warm seclusions (Shapiro-Keyser cyclones) using an image segmentation approach; the development of a set of 33 notable storms, used to refine the SJP diagnostic, holistically assessed for likely sting-jet

presence (using observations, previous literature and detailed analysis of the location and timing of the regions with CSI); and comprehensive comparison of the characteristics and composite structure of cyclones with and without the SJP, mainly through the use of ERA5 data but also using an observational product: a blend of scatterometer observations with bias-corrected ERA5 data.

The remainder of this paper is structured as follows. The reanalysis data and cyclones tracking algorithm used are described

in Sect. 2 followed by the warm-seclusion algorithm, satellite data, SJP diagnostic, set of notable storms, and cyclone compositing method. The results section (Sect. 3) begins with a short analysis of how the proportion of warm-seclusion cyclones varies with cyclone intensity before a detailed analysis of how the frequency, spatial and temporal variability, intensity metrics, and composite structure of cyclones with and without the SJP compare, before finishing with a short analysis of the blended scatterometer data. The conclusions section (Sect. 4) finishes the paper.

## 2  Methods

### 2.1  Reanalysis data and cyclone tracking

Cyclones were identified using the ERA5 dataset, which is the latest comprehensive global reanalysis dataset produced by the European Centre for Medium-Range Weather Forecasts (ECMWF). ERA5 was produced using cycle Cy41r2 of ECMWF's integrated forecast system (IFS), which was operational from 8 March to 21 November 2016. In this cycle the IFS model had a





horizontal spectral truncation of TL639 with 137 terrain-following hybrid-pressure levels up to a height of 80 km. Atmospheric data is available on a regular latitude-longitude grid with $0.25°$ ($\sim30$ km) spacing. For this study cyclones were identified and tracked using hourly ERA5 data for 43 years of extratropical (defined as $40$–$80°$) extended winters in the Northern Hemisphere (October–March) from 1979–80 to 2021–22 and in the Southern Hemisphere (April–September) from 1980 to 2022.

The TRACK algorithm (version 1.5.2) was used to track the cyclones (Hodges, 1995, 1999, 2023). The cyclones were identified as maxima of spatially smoothed 850-hPa relative vorticity ($\xi_{850}$) fields with the smoothing performed using spectral filtering to retain features with total wave numbers between T5 and T63 (where T refers to the triangular truncation of the spherical harmonics), which equates to an approximately smallest resolved scale of 320 km for circular features. Cyclone features with maximum smoothed $\xi_{850}$ values exceeding $10^{-5}$ s$^{-1}$ were retained and tracked with the tracks retained if they lasted for more than one day with a total track distance exceeding 500 km; this duration was chosen to retain all of the the cyclones in the notable storms list used for verification. Spatial statistics such as track and genesis density were computed from the tracks using spherical kernels (Hodges, 1996).

After calculating the tracks using the hourly ERA5 data (as higher temporal resolution data improves the track accuracy), the temporal resolution of the track files was then degraded to the six-hourly times (00, 06, 12, and 18 UTC only) at which the sting-jet precursor (SJP) was evaluated. To keep the number of tracks manageable for further processing, we choose to focus on the top decile of tracks in terms of the maximum intensity (defined as the maximum value of $\xi_{850}$) along the extratropical portion of the track and to consider only cyclones in which a warm seclusion is present at the time of maximum intensity (as sting jets have only been identified as descending towards the surface in such Shapiro-Keyser cyclones). Percentiles were computed separately for each hemisphere and the 90$^{\text{th}}$ percentile values for spatially smoothed $\xi_{850}$ are $12.72 \times 10^{-5}$s$^{-1}$ and $13.95 \times 10^{-5}$s$^{-1}$ in the Northern and Southern Hemispheres, respectively. Note that the Northern Hemisphere absolute threshold values were also used for the North Atlantic and North Pacific regions. These thresholds equate to local percentiles of 86% for both regions implying that cyclones in these basins are typically more intense than in the Northern Hemisphere overall. After sub-selecting cyclones in which a warm seclusion was diagnosed (as described in Sect. 2.2), the SJP diagnostic (Sect. 2.4) was applied to sets of 5550 Northern Hemisphere and 5207 Southern Hemisphere cyclones. For each cyclone the fields required for the SJP diagnostic were extracted from the ERA5 dataset in a square domain of $\pm15°$ in both latitude and longitude (with a spherical cap projection) from each track point from 42 h before to 42 h after the time of maximum intensity. To enable comparison with previous studies that considered all cyclones in the North Atlantic, a randomly selected year of data was also processed for all cyclones in that year.

## 2.2 Identification of warm-seclusion cyclones

The maps in Fig. 1 illustrate the algorithm used for warm-seclusion cyclone identification. Three examples are shown: two cyclones identified as having a warm seclusion (one from each hemisphere) and one that failed the identification test. (Note that the two warm-seclusion cyclones are in the notable storms set described in Section 2.5.) The two panels shown for each cyclone show the low-level cyclone structure (mean-sea-level pressure (MSLP) field with the identified cyclone core together





with the 850-hPa wet-bulb potential temperature ($\theta_w$) field) and warm-seclusion diagnostic process, respectively. A short description of the algorithm follows.

Warm-seclusion identification is only applied to a track if the associated cyclone centre enters the mid-latitudes (i.e., the MSLP cyclone centre associated with the tracked vorticity feature is for at least one six-hourly timestep within 40° and 80° of latitude in either hemisphere). The time chosen for the performing the identification is that of cyclone maximum intensity (when $\xi_{850}$ is maximum along the track) while the cyclone centre is in the mid-latitudes. The minimum value of MSLP at the time of maximum intensity in a 5° radius (using a geodesic great circle radius, B-spline interpolation and steepest descent

minimisation) centred on the track point is identified as the cyclone centre. Then, the pressure core is defined as the region where MSLP is less than 4 hPa above its cyclone-centre minimum and less than 300 km away from it.

    A type of image segmentation algorithm called a watershed algorithm is then run on the 850-hPa $\theta_w$ to identify distinct watersheds within the warm sector (using a footprint size of $8 \times 8$ grid points). The warm sector is defined as the region where 850-hPa $\theta_w$ is warmer than its maximum value in the cyclone core minus 2 K. If the watershed that overlaps the cyclone

pressure core with the most points is all contained within a 600 km-radius circle centred at the cyclone centre, the cyclone is identified as having a warm seclusion (provided this watershed exceeds a threshold size of 25 grid points). In the right-hand-side maps in Fig. 1 the watershed boundary is highlighted with a thick green contour if a warm seclusion has been identified. Otherwise, the cyclone is identified as a non-warm-seclusion cyclone, and the watershed boundary is highlighted with a thick red contour. Our warm seclusion identification method includes several user-defined thresholds. The values for these thresholds

were optimised by analysing the behaviour of the algorithm for the set of notable storms as well as other cyclones to provide the best agreement between subjective and algorithm-based warm seclusion identification. The starting point for our algorithm development was the algorithm used to identify cyclones with warm seclusions by Manning et al. (2021). Their algorithm identifies warm seclusions as localised warm regions, where 850-hPa $\theta_w$ is at least 2 K warmer than the surroundings, that overlap with a MSLP core defined as points where the MSLP is within 6 hPa of the minimum values. The most important

difference to the Manning et al. (2021) approach is that, partly to increase efficiency, we have used a watershed algorithm to identify the warm core rather than identifying a warm peak by searching in turn along each latitudinal row and longitudinal column of the model data. We also used image processing to identify the cyclone MSLP core, whereas Manning et al. (2021) used the same line-by-line search to identify the cyclone core.

### 2.3   Observational data

We utilise several independent sources of satellite data in this study as input into the expert judgement evaluation process for the set of notable storms outlined in Sect. 22.5 and to compare the sets of cyclones with and without a diagnosed SJP.

    Passive radiometer images were obtained from three sources: the High Rate SEVIRI IR10.8 band from the MSG 0° satellite, which covers the Atlantic sector and is available at 15 min frequency from 2004-present; the Advanced Very High Resolution Radiometer (AVHRR) IR10.8 band from the METOP satellites, which are polar orbiters and therefore available on daily swaths

covering limited areas; and the Geostationary Ring IR10.8 Multimission Image produced by EUMETSAT, which merges three-hourly images from several geostationary satellites to provide global coverage. In each case, we use the satellite image



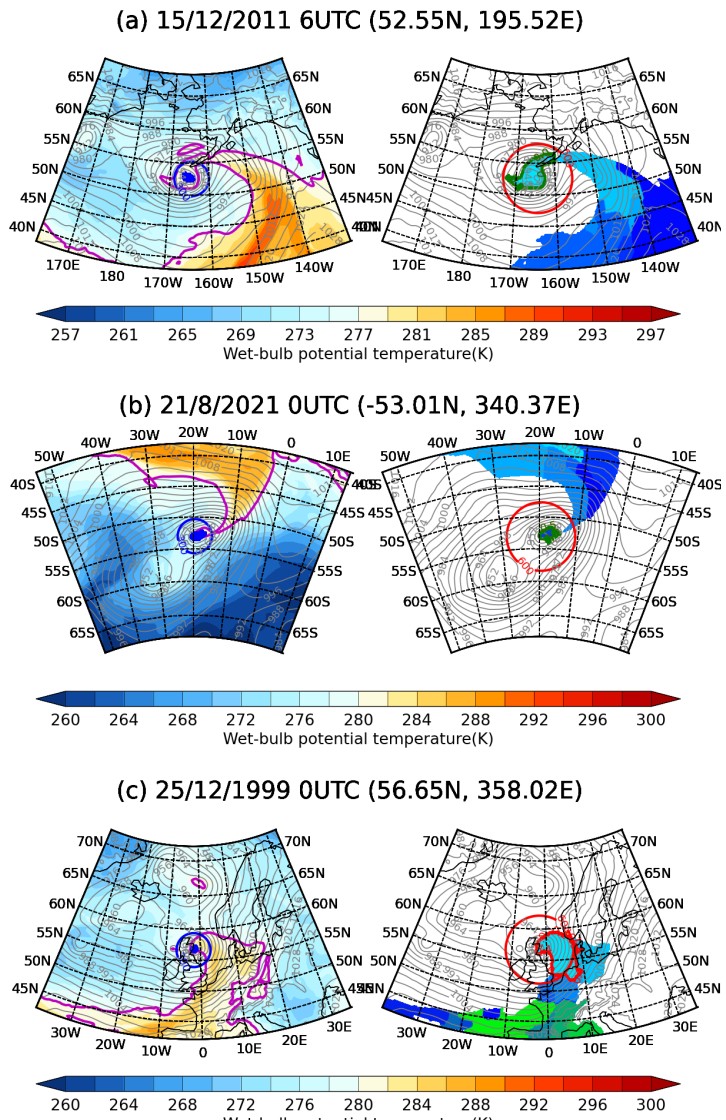

**Figure 1.** Examples of maps used for evaluation of the warm-seclusion identification algorithm. Left-hand-side maps include MSLP (grey contours, hPa), 850-hPa $\theta_w$ (shading, every 1 K) with warm-sector areas (see definition in the text) highlighted by thick magenta contour. The blue circle (cyclone pressure core) has a radius of 300 km and is centred over the cyclone centre. The blue-shaded region covers the cyclone pressure core. Right-hand-side maps include MSLP (grey contours, hPa) and all warm-sector watersheds identified by the algorithm (discrete colours). The thick green or red contour highlights the watershed that overlaps the pressure core with the most points and its colour depends on whether the cyclone is identified as having a warm seclusion (green) or not (red). The red circle for warm-seclusion identification has a radius of 600 km and is centred over cyclone centre.



closest in time to that of the maximum intensity of the cyclone, favouring the higher-resolution AVHRR images over the SEVIRI/Multimission images when they are available. Example AVHRR images are shown in Fig. 2 for two of the notable storms.

Ocean surface wind data are obtained from the Cross-Calibrated Multiplatform Version 3.0 dataset (CCMP3.0; Mears et al., 2022a) which provides gridded vector 10-m wind data over ocean points on a $0.25 \times 0.25°$ grid from 1993–2019. CCMP3.0 uses a variational analysis method to blend ocean surface wind retrievals from a wide range of satellite microwave sensors with a background field of adjusted 10-m neutral winds from ERA5 to produce a regular six-hourly dateset (Atlas et al., 2011). The ERA5 adjustment involves removing ocean currents vectors to transform to obtain wind speeds relative to the ocean surface,
and applying a quantile-mapping technique to bias correct strong wind values which are typically too weak in ERA5. The data is available at six-hourly intervals from 1993–2019. Examples of CCMP data are also shown in Fig. 2.

### (a) 15 Dec 2011 06UTC (52.55N, 195.52E)

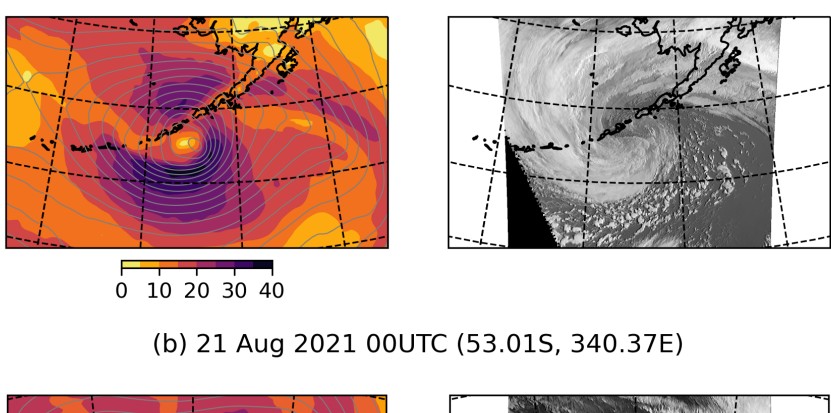

### (b) 21 Aug 2021 00UTC (53.01S, 340.37E)

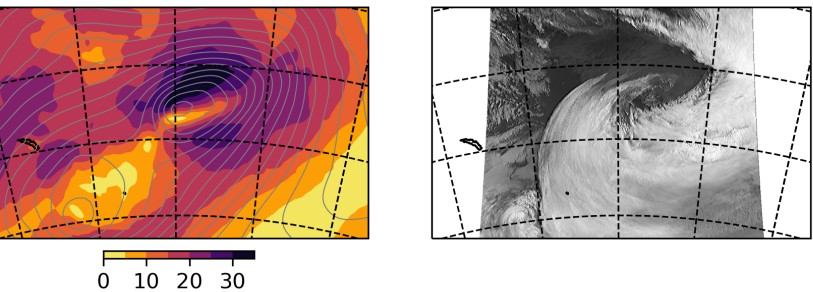

**Figure 2.** Satellite observations of the example storms from Figs 1a and b. Left hand panels show 10-m wind speed values (units: $\mathrm{m\,s^{-1}}$) from the CCMP3.0 blended wind product at the times indicated together with contours of MSLP from ERA5 (interval: 4 hPa), and right hand panels show IR10.8 band images from the AVHRR instrument for the image that is closest in time to the time shown. In both cases the data are re-projected onto the same projection used in Fig 1, but note the area shown is more zoomed in than in Fig 1. For full details of the dataset used, see Sect. 22.3.




## 2.4 Sting-jet precursor diagnosis including calibration and verification across ocean basins

The SJP diagnostic, based on the methodology outlined in Martínez-Alvarado et al. (2013), is applied to the cyclone data from 12 h before to 6 h after the time of cyclone maximum intensity to focus on the time period when sting jets have been found most likely to occur in case studies. The SJP diagnostic uses a metric for CSI, specifically the energy available to downdraught air through the release of CSI termed the downdraught slantwise convective available potential energy (DSCAPE) (Gray et al., 2011). The CSI assessment is then complemented by a set of additional criteria to ascertain that the CSI is within an environment conducive to the formation of sting jets. These criteria consist of an evaluation of the availability of moisture for CSI release and the location of CSI relative to the cyclone centre, the bent-back front and the cloud head. Model grid points meeting the threshold DSCAPE value and environmental criteria are hereafter referred to as "CSI points". This methodology has been used, with variations, in climatological assessments (Martínez-Alvarado et al., 2012; Hart et al., 2017), climate change studies (Martinez-Alvarado et al., 2018), and in near-real-time forecasting operations (Gray et al., 2021).

The precursor diagnostic settings used are the same as those in Gray et al. (2021) with the following modifications to the location of CSI. For CSI release to be considered as having the potential to generate sting jets, CSI must be located in the cyclone's cloud head. Therefore, any CSI diagnosed in a sector (of a circle centred on the cyclone) associated with the cyclone's warm conveyor belt must be disregarded. In Gray et al. (2021) the warm conveyor belt sector was defined as an asymmetric sector between $60°$ and $-100°$, with these angles measured counterclockwise with respect to the cyclone's direction of travel. Here this sector is symmetric with respect to this direction, defined by an angle $2\alpha$, which was set to $240°$ i.e., spanning from $120°$ and $-120°$.

The cyclone's travel direction is computed at each track point based on the location of the cyclone's centre, using a forward-difference approach for the first time step, a backward-difference approach for the last time step, and a centred-difference approach for any other time step in the cyclone's track. For the forward and backward differencing the time interval between the chosen track points was 6 h. The centred-difference method was implemented using track points from 12 h before to 12 h after the time of the track point of interest except for the second and second to last time steps when 6 h was used. The longer time interval was preferred where possible because this improved the smoothness of the travel direction's evolution. This implementation differs from Gray et al. (2021), in which centred differencing was used for all time steps, but the first and last time steps were discarded.

The cloud head sector assessment assumes a typical eastward and poleward cyclone travel direction. Therefore, an additional requirement was introduced in this work to flag cyclones with atypical tracks. On inspection, these cyclones were sometimes those close to maturity tracking slowly with a strong poleward component to their track and occasionally those earlier in their lifecycle performing small track loops meaning that for a short period the travel direction had a strong westwards or equatorward component. Measuring the angles with respect to north (such that east is $90°$), a cyclone track was considered atypical if the cyclone's travel direction was between $180°$ and $315°$ in the Northern Hemisphere and between $225°$ and $360°$ in the Southern Hemisphere at any time when the SJP diagnostic was calculated.





To classify the cyclones according to their likelihood of containing a sting jet (in reality), the total number of CSI points diagnosed for each cyclone is considered. The total (rather than, e.g., an instantaneous maximum) is used because there is uncertainty about when CSI appears in a low-resolution model and so the exact timing is not relevant as long as it is around the time of maximum intensity. The cyclones have been classified into three categories according to the output of the SJP diagnostic: sufficient CSI so likely to produce sting jets (TRUE, $> 16$ CSI points), insufficient CSI so unlikely to produce sting
jets (FALSE, $< 9$ CSI points), and intermediate CSI so neither likely or unlikely to produce sting jets (MARGINAL, $9 \leq$ CSI points $\leq 16$). We assessed and refined the methodology using the set of notable storms described in Section 2.5. In particular, the results were found to be largely insensitive to the precise values used for $\alpha$ and the time interval used to calculate the cyclone travel direction and, while varying the threshold numbers of CSI points led to changes in the classification of some cyclones, these changes were found to be relatively small. The output of the sting-jet precursor algorithm is illustrated in Fig. 3,
corresponding to the storms in Fig. 2, showing that it is correctly identifying instability in the cloud head tip.

## 2.5 Selection of notable storms

A set of 33 "notable storms" was used to test and refine the algorithm before applying it to the complete cyclone datasets. These were intense cyclones generally associated with severe impacts and which in some cases were the subject of published case studies. These notable storms were all identified as $>90$th percentile in intensity, warm-seclusion cyclones (see Sect. 2.2)
and included cases from the North Atlantic (both near the North American East coast and further east all the way to western Europe), North Pacific (including the North American West Coast) and southern ocean basins. The notable storms are listed in Tables S1 and S2 included in the Supplementary Material.

  Where present, satellite imagery and ocean surface wind data (see details in Sect. 2.3), together with published results where available, were analysed. Output from the SJP diagnostic (Sect.2.4) was also examined qualitatively, focusing on the relative
locations of cloud head sector and CSI points, the direction of travel of the storms and the number of CSI points present (using map-style plots such as those in Fig. 3). All these data were used to form an "expert judgement" (on a five-point scale: TRUE, LIKELY TRUE, MARGINAL, LIKELY FALSE, FALSE) of the likelihood of sting jet presence (see scores and brief score justifications in Table S2). This manual evaluation of the storm likelihood to contain a sting jet was then compared against the automatic SJP score (TRUE, MARGINAL or FALSE). There was satisfactory agreement between the two sets of scores as they
were equivalent or near-to-equivalent (such that the manual and SJP assessments were, respectively, TRUE-TRUE, LIKELY TRUE-TRUE, MARGINAL-MARGINAL, LIKELY FALSE-FALSE, or FALSE-FALSE) in 22 cases out of 33, while 9 of the remaining 11 involved a MARGINAL score in one of the two sets, indicating that either the manually analysed evidence was not clear enough for a conclusive verdict or that the results from the precursor algorithm did not give a clear indication. The only two cases of clear disagreement (storm Martin and the October 2021 Nor'easter storm, both matched as LIKELY FALSE-
TRUE) are associated with a lack of clear satellite imagery and a cyclone with a complex track, respectively, and are thus cases where considerable uncertainty applies to either the manual or automatic classifications. Note that the large fraction of TRUE cyclones, 22 according to the SJP diagnostic compared to 6 classified as FALSE and 5 as MARGINAL, does not indicate that

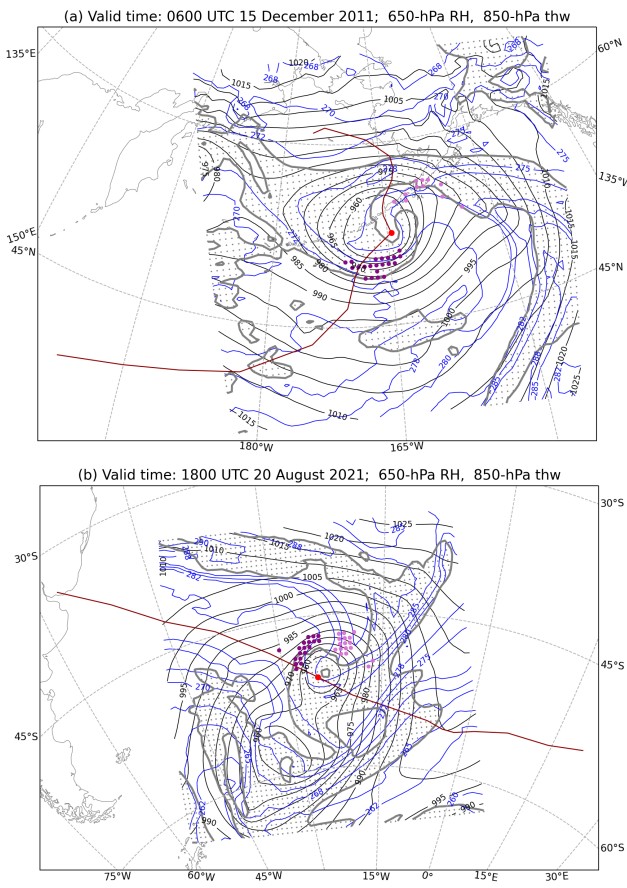

**Figure 3.** Illustration of the implementation of the SJP diagnostic, showing MSLP (black contours, hPa), 850-hPa $\theta_w$ (blue contours, K), and 650-hPa relative humidity with respect to ice above 80% (grey contour and stippling) at (a) 0600 UTC on 15 December 2011 and (b) 1800 UTC on 20 August 2021. The red lines represent cyclone tracks with the red circles marking the cyclone centres at the validation times. Purple and pink dots represent CSI points satisfying and not satisfying the additional location criteria, respectively.

the results of the algorithm are skewed towards TRUE, as several of the notable storms are known or hypothesised sting-jet cyclones (with at least seven of them documented as such in the literature).

## 2.6 Cyclone composites


The structure of the cyclones with and without the SJP (excluding marginal cases) have been compared through compositing cyclone fields according to the time relative to the time of maximum cyclone intensity and the ocean basin (Northern or Southern Hemisphere, North Atlantic or North Pacific). The North Atlantic and North Pacific basins are defined throughout




this study as being between the longitudes of 80°W–30°E and 120°E–120°W, respectively[1]. The cyclones are rotated prior to
compositing such that their travel direction points eastward with the aim of better aligning the major cyclone features, such as
fronts, between the cyclones. Cyclones with atypical travel directions, defined as in Sect. 2.4, are excluded from the composites
(at the time that the direction is anomalous only), as after rotation the orientation of their major features tends to be anomalous.

Cyclone relative fields are extracted for the SJP calculation gridded using a radial coordinate system centred on the MSLP
centre of each cyclone and an additional field, potential vorticity, was extracted in the same way for compositing. As described
in Sect. 2.2, the extracted domain is a square that extends ±15° from the storm centre. After compositing the domain was
reduced to ±12° from the storm centre to exclude the outer portions of the domain where the fields were more noticeably noisy
due to missing data for some of the cyclones after rotation, though note that there will still be slightly fewer data points in the
composites near the edges of the domain shown, especially in the corners.

## 3 Results

### 3.1 Characteristics of warm-seclusion cyclones

As the SJP diagnostic was only applied to those cyclones exceeding the 90th percentile in maximum intensity and that have
a diagnosed warm seclusion, the proportion of cyclones with a warm seclusion as a function of intensity decile is presented
in Fig. 4 to show the effect of the warm-seclusion constraint. There is a clear, more-than-linear increase in the fraction of cy-
clones that are identified as having warm seclusions as their maximum intensity increases, with the fraction of warm-seclusion
cyclones being similar for both hemispheres and the three major oceanic basins. The fraction of warm-seclusion cyclones is
0.42–0.45 for cyclones in the lowest intensity decile increasing to 0.73–0.77 for cyclones in the highest intensity decile such
that the vast majority of cyclones in the top decile are diagnosed as having a warm seclusion. Consequently, the warm-seclusion
constraint reduced the number of cyclones evaluated using the SJP diagnostic by less than 30% although, due to the computa-
tional expense of applying the algorithm, this reduction was still considered worthwhile (and greater benefit would be obtained
if applying the SJP to a broader cyclone intensity range).

### 3.2 Frequency of cyclone types

The absolute and relative frequencies of cyclones with and without the SJP (termed SJP and nSJP cyclones, respectively)
are shown in Table 1. Of the more than 10,000 warm-seclusion cyclones evaluated, 29% had a SJP and 55% did not have
a SJP (with the remainder categorised as marginal). Assuming that a sting jet would not have been able to descend towards
the surface in all the cyclones without a warm seclusion (irrespective of the presence of a sting-jet precursor), 21% of all top
intensity decile cyclones may have had a sting jet. Cyclones in the Southern Hemisphere are only just over half as likely to
have the SJP as those in the Northern Hemisphere (20% of evaluated cyclones compared to 37%). The percentage of cyclones

---

[1]For compositing the location of the tracked vorticity centre was considered for convenience rather than its associated MSLP centre (as used in the rest of
the analysis)





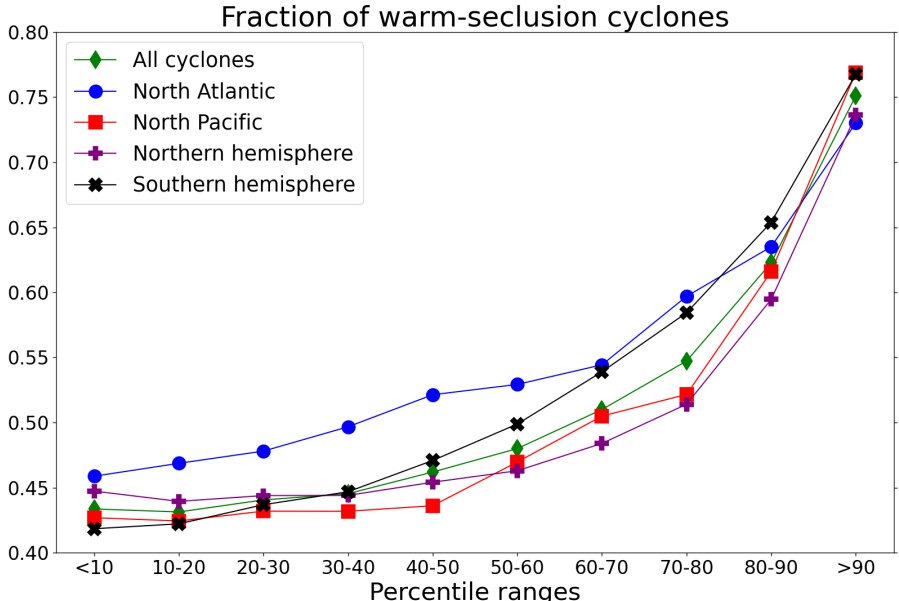

**Figure 4.** Fraction of warm-seclusion cyclones for the full dataset and for hemispheric and main oceanic subsets. Intensity deciles are based on maximum 850-hPa $\xi_{850}$ (see Sect. 2.1) and are calculated separately for each hemisphere (but not for each oceanic basin).

in each precursor category was similar for North Atlantic and North Pacific basins and when considering the whole Northern Hemisphere.

The percentage of SJP cyclones found for the North Atlantic in this 43-year climatology based on ERA5 data (38% of evaluated cyclones) is similar to that found by Hart et al. (2017) for this region using 32 years of data from the preceding ERA-Interim reanalysis for the same extended winter season and using a very similar SJP diagnostic. They found that 32% of the 5447 cyclones tracked had the precursor; an almost identical percentage (33%) was also obtained from analysis of a climate model simulation of the present-day climate by Martinez-Alvarado et al. (2018). However, our results are not directly

comparable due to differences in the selection of cyclones to evaluate and in the methodology. In particular, Hart et al. (2017) evaluated all cyclones lasting at least two days with total track distance exceeding 1000 km, whereas here only warm-seclusion cyclones in the top intensity decile lasting at least one day and with track distance exceeding 500 km were evaluated.

To assess the consistency between the current and previous results applying the SJP diagnostic to North Atlantic cyclones and provide indicative results for how the percentage of SJP cyclones varies with cyclone intensity decile, all tracked cyclones

(all intensities and both with and without a warm seclusion) with minimum track durations of both one and two days (and associated minimum track distances of 500 and 1000 km, respectively) were evaluated for this ocean basin for a single winter season. The 2011–2012 season chosen is representative of the 43-winter climatology in terms of the percentages of top intensity decile North Atlantic warm-seclusion cyclones found in the SJP, nSJP and marginal categories for the same one-day minimum track duration (Table 1). This season was also included in the period evaluated by Hart et al. (2017) and contains windstorm



Friedhelm from the set of notable storms. Comparison of the top decile 2011–12 winter North Atlantic cyclones with all North Atlantic cyclones from that winter (with the same track constraints) reveals SJP cyclones occur about twice as frequently in the former providing strong evidence that SJPs are preferentially found in more intense cyclones, consistent with Hart et al. (2017) when comparing all cyclones with explosively developing cyclones. Recall that the proportions of cyclones in the top intensity decile subsets exceed 10% in the North Atlantic and North Pacific subregions because the intensity threshold used

is an absolute value calculated using cyclones in the entire Northern Hemisphere. Increasing the minimum track duration increases the proportion of cyclones found in the North Atlantic subset, implying that in general the shorter duration cyclones are weaker. The proportions of SJP and nSJP top decile warm-seclusion cyclones are similar for a given track constraint, again implying a strong relationship between (absolute) maximum intensity and SJP likelihood.

The SJP is also found to be present in 2011–12 cyclones that are not diagnosed as having a warm seclusion. Indeed, for ex-

ample, the proportions of SJP and nSJP cyclones for all-intensity non-warm-seclusion cyclones with a minimum track duration of one day are 18 and 74%, respectively (with the remainder being marginal cyclones), and those for the top decile subset are both 44%. Hence, the warm-seclusion and non-warm-seclusion cyclones are similarly likely to be SJP cyclones (e.g., 17% of all warm-seclusion cyclones have a SJP from Table 1). Conversely, similar proportions of SJP and nSJP cyclones have a warm seclusion (57 and 58%, respectively, for all-intensity cyclones). So, warm seclusion cyclones are not more likely to have the

SJP compared to cyclones without a warm seclusion, and SJP cyclones are not more likely to have a warm seclusion compared to nSJP cyclones. More detailed analysis of the SJP non-warm-seclusion top decile cyclones (not shown) has confirmed the existence of CSI points in the cloud head tip (as required to diagnose the SJP) in some cyclones that have been correctly diagnosed as not having a warm seclusion. To date in published studies, sting jets have only been diagnosed through detailed analysis in cyclones following the Shapiro-Keyser conceptual model evolution, with a frontal fracture and a warm seclusion

at maturity. The existence of the SJP in some cyclones that do not follow this evolution suggests that, while the type of instability commonly found in sting-jet cyclones can develop, the absence of a frontal-fracture region with the associated weak stability inhibits the descent of the sting jet towards the ground and consequently its potentially damaging impact. Hence, in determining whether a cyclone may yield sting jets cyclone, structure should be considered together with cloud head instability characteristics. However, more research is needed to needed to determine whether sting jets could occur in cyclones that do

not follow the Shapiro-Keyser evolution.

When considering all minimum two-day track duration 2011–12 cyclones (all intensities and both with and without a warm-seclusion), the percentage of SJP, nSJP and marginal cyclones are 21, 68, and 11%, respectively, with the sum of SJP and marginal cyclones yielding a percentage virtually identical to that of SJP cyclones found by Hart et al. (2017) and Martinez-Alvarado et al. (2018) for the same track constraints. Hence, our results for the North Atlantic basin are consistent with those

of earlier studies using the same track constraints despite the use of the newer ERA5 reanalysis and changes made to the SJP diagnostic. However, they suggest that the majority of the cyclones categorised as marginal precursor cyclones here would instead have been categorised as SJP cyclones in these earlier two-category precursor studies. For the remainder of this paper we only consider the results from the 43-winter climatology and focus on comparison between the SJP and nSJP cyclones, so ignoring the marginal cases, to emphasize the differences between cyclones that do and do not have the SJP.





**Table 1.** Absolute and relative (%) frequency of SJP, nSJP, and marginal SJP warm-seclusion cyclones and non-warm-seclusion cyclones (with minimum one day track duration) by hemisphere, major oceanic basin and overall for the 43-winter climatology, and the 2011–2012 frequencies for the North Atlantic for two choices of minimum track duration (in all cases, the one and two day minimum track durations are associated with minimum total track distances of 500 and 1000 km respectively). The top decile intensity thresholds are calculated separately for each hemisphere (but not for each oceanic basin) and consequently a larger proportion of cyclones are considered in the North Atlantic and North Pacific subregions than in the two hemispheres. The values in brackets are the percentages of all cyclones and of warm-seclusion cyclones only, respectively. Note that the percentage of marginal and non-warm-seclusion cyclones are fairly consistent across region: (11–14%, 15–18%) and (23–27%, 30–37%), respectively.

| Region | Constraints | | winters | warm-seclusion cyclones | | | non-warm-secl. | Total |
|---|---|---|---|---|---|---|---|---|
| | max. $\xi_{850}$ | length/d | | SJP | nSJP | marginal | | |
| Overall | top 10% | 1 | ×43 | 3068 (21, 29%) | 5967 (42, 55%) | 1722 | 3565 | 14322 |
| N. Hemi. | top 10% | 1 | ×43 | 2040 (27, 37%) | 2541 (34, 46%) | 969 | 1989 | 7539 |
| S. Hemi. | top 10% | 1 | ×43 | 1028 (15, 20%) | 3426 (51, 66%) | 753 | 1576 | 6783 |
| N. Pacific | top 14% | 1 | ×43 | 1018 (27, 35%) | 1343 (36, 47%) | 510 | 863 | 3734 |
| N. Atlantic | top 14% | 1 | ×43 | 891 (27, 38%) | 1061 (33, 45%) | 421 | 877 | 3250 |
| N. Atlantic | top 16% | 1 | 2011–12 | 20 (24, 36%) | 26 (31, 46%) | 10 | 27 | 83 |
| N. Atlantic | top 28% | 2 | 2011–12 | 18 (24, 36%) | 22 (30, 44%) | 10 | 24 | 74 |
| N. Atlantic | all cyclones | 1 | 2011–12 | 51 (10, 17%) | 222 (43, 74%) | 29 | 215 | 517 |
| N. Atlantic | all cyclones | 2 | 2011–12 | 33 (12, 20%) | 110 (42, 67%) | 20 | 102 | 265 |

### 3.3 Spatial and temporal variability SJP and nSJP cyclones

The spatial distributions of SJP and nSJP cyclones are compared in Fig. 5. Considering first the track densities in the Northern Hemisphere (Fig. 5(a)), the location of the maximum track density of the SJP cyclones is found to the southwest of that of the nSJP cyclones in both the North Atlantic and North Pacific ocean basins. Hence, SJP cyclones travel at lower latitudes and are more frequent in the early part of the storm tracks. Despite the differences in track density structure between the SJP

and nSJP cyclones, the maximum densities are fairly similar for both ocean basins. In the Southern Hemisphere the storm tracks are more zonal (Fig. 5(c)). The maximum track density for the SJP cyclones occurs to the north of that for the nSJP cyclones (i.e., SJP cyclones have a tendency to track at more equatorward latitudes in both hemispheres) although, unlike in the Northern Hemisphere, there is no obvious longitudinal displacement. Also unlike in the Northern Hemisphere, nSJP cyclones have substantially higher densities than SJP cyclones, with the peak value approximately double (consistent with the far fewer

SJP compared to nSJP cyclones here, Table 1).

As in previous cyclone tracking studies (e.g., Hoskins and Hodges, 2002), localised "hot spots" of enhanced genesis densities are found. In the Northern Hemisphere (Fig. 5(b)) these occur over the Gulf Stream and Kuroshio currents for both the SJP and nSJP cyclones although in both locations the genesis density peak for the SJP cyclones is slightly to the south of that for



the nSJP cyclones and the peak densities for the SJP cyclones are higher than those for the nSJP cyclones, especially for the

Kuroshio current region. A hot spot of enhanced genesis density is also apparent over the Rocky mountains, but only for the

SJP cyclones, and additional hot spots are found for the nSJP cyclones towards the eastern ends of the storm tracks (regions

typically associated with secondary cyclogenesis). In the Southern Hemisphere (Fig. 5(d)) the hot spots are scattered along the

Southern Hemisphere storm track with the SJP cyclone hot spots found equatorward of those of the nSJP cyclones. Particularly

notable is the enhanced genesis density over South America which peaks more than 15° equatorward for the SJP compared to

the nSJP cyclones.

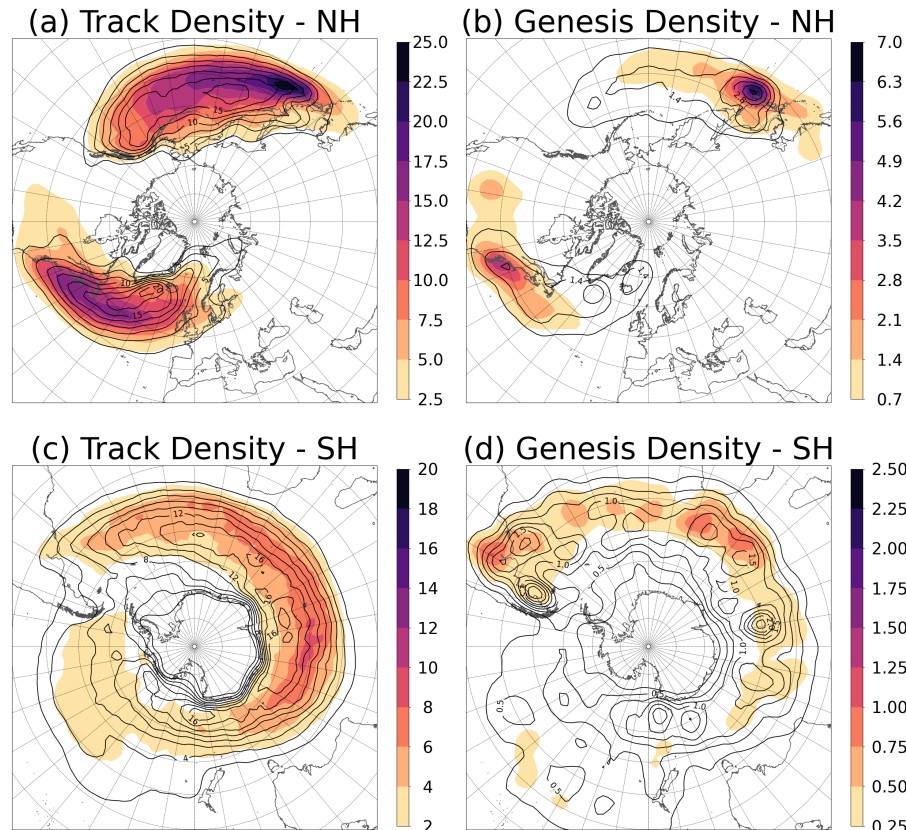

**Figure 5.** Track (a,c) and genesis (b,d) densities for SJP (shading) and nSJP (black contours) cyclones for the (a-b) Northern and (c-d) Southern Hemisphere. Units are the number per unit area per season where the unit area is equivalent to a 5° spherical cap (approximately $10^6$ km$^2$). Shading and contours share the same intervals in each of the panels.

The temporal variability of the cyclones is presented in Fig. 6, beginning with the monthly cyclone frequency over the

six-month extended-winter season considered (Fig. 6(a)). The monthly cyclone frequency has different behaviour for the SJP



and nSJP cyclones. In the Northern Hemisphere (and in both the North Atlantic and North Pacific basins separately) there is a strong monthly cycle in SJP cyclones with a peak in mid winter (December and January). In contrast, there is a more uniform

SJP cyclone frequency in the Southern Hemisphere. For the nSJP cyclones there is no consistent frequency variability between the Northern Hemisphere, North Atlantic and North Pacific basins and in the Southern Hemisphere there is a peak towards the later winter months (July and August). The inter-annual variability is shown in terms of the number of cyclones and the percentage of SJP cyclones in Figs. 6(b) and (c), respectively. There are positive trends in cyclone numbers for all cyclone categories throughout the evaluation period, with the most robust trends being for the SJP cyclones in the North Atlantic and

Northern Hemisphere and for both the SJP and nSJP cyclones in the Southern Hemisphere. Since only cyclones in the top intensity decile have been evaluated, positive trends in the number of cyclones evaluated for the SJP either reflect an increase in the total number of cyclones or in the intensities of cyclones (such that more cyclones in later years are found in the the top intensity decile). The percentage of cyclones that have the SJP provides an alternative metric for inter-annual cyclone variability. There are positive, but not significant, trends in this metric for each hemisphere and basin.

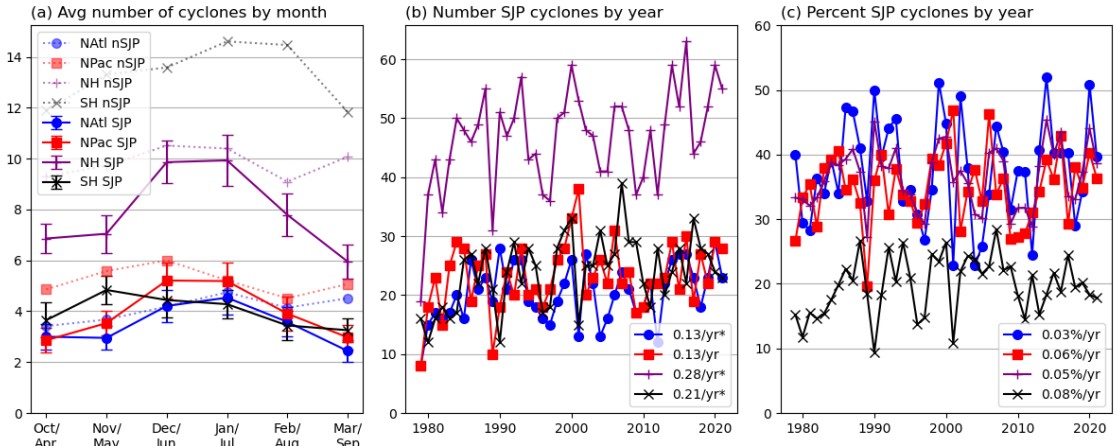

**Figure 6.** Monthly (a) and inter-annual (b,c) variations of SJP cyclone counts. Units in (a) are average number of cyclones per month, in (b) are absolute cyclone counts, and in (c) are the percentage of all cyclones that are SJP cyclones. The error bars in panel (a) (for legibility, shown for SJP cyclones only) indicate two standard errors in the means estimated from the 43 yearly values for each month, and the legends in panels (b) and (c) indicate the linear trends of each timeseries with astericks indicating trends that are non-zero according to a two-sided Wald Test at the 5% level.

**3.4 Distributions of intensity metrics of SJP and nSJP cyclones**

Further evidence for the distinctiveness of cyclones with and without the SJP is now provided by comparing the histograms of the following cyclone intensity metrics for the Northern and Southern Hemisphere cyclones (Fig. 7): maximum $\xi_{850}$, maximum low-level (925-hPa) wind speed, minimum MSLP and a simple metric for MSLP anomaly (maximum MSLP difference within 300 km of the minimum MSLP) all at the time of maximum intensity (according to $\xi_{850}$), and the maximum 24-h MSLP



decrease along the track. For all metrics, except for the minimum MSLP at the time of maximum intensity in the Southern Hemisphere, there is a clear distinction between the normalised frequency distributions with the SJP cyclones being more intense (i.e., higher values of maximum $\xi_{850}$, low-level windspeed, MSLP anomaly and 24-hPa decrease, and lower values of minimum MSLP). This distinction is visible in the shapes of the SJP and nSJP cyclone distributions and also in the mean values which increase in intensity going from the set of nSJP (FALSE) cyclones to that of marginal cyclones and then to SJP

(TRUE) cyclones. For the maximum $\xi_{850}$, minimum MSLP and MSLP anomaly the higher intensity tail is also longer for SJP cyclones with this long tail particularly notable for the maximum $\xi_{850}$ distribution in the Northern Hemisphere. Comparing the distributions for the two hemispheres, the mean values of the Southern Hemisphere cyclones for the three categories are slightly higher for maximum $\xi_{850}$, low-level wind speed and 24-hPa MSLP decrease. However, the more notable difference is for minimum MSLP, with Southern Hemisphere mean values for the three categories being 10–18 hPa lower than for their

Northern Hemisphere counterparts. The apparently anomalous result that there is not a clear distinction between the frequency distributions of minimum MSLP for the SJP and nSJP cyclones in the Southern Hemisphere is a consequence of the difference in their genesis and track densities (Fig. 5) with SJP cyclones typically forming closer to the equator (where climatological MSLP is higher) and travelling more zonally for most of their life cycle. The along-track evolution of the minimum MSLP is further discussed in the following section on composite analysis (Sect. 33.5).

**3.5 Composite SJP and nSJP cyclone characteristics**

Cyclone compositing reveals systematically distinctive characteristics of cyclones with and without a SJP. Maps of selected composite fields at chosen times relative to the time of maximum cyclone intensity and for chosen ocean basins are shown in Figs. 8–11 and S1–S5, and line plots showing time series of attributes of some of these fields (e.g., composite maximum 850-hPa wind speed) are shown in Fig. 12 for all four regions and cyclones with and without the SJP. Recall when compass

directions are used for convenience below that the cyclones have been rotated prior to compositing. While the compositing, by its nature, yields smoothed cyclone fields, the key cyclone features including the cloud head, fronts, warm sector and deep low-pressure centre can clearly be identified in the map plots. At the time of maximum intensity the composite SJP cyclone has notably stronger low-level winds and warmer (in $\theta_w$) warm sector and cyclone core than the composite nSJP cyclone in all three regions (North Atlantic, North Pacific and Southern Hemisphere) shown in Fig. 8, which shows composites for the SJP

cyclones in the top row and composite differences (SJP minus no nSJP) in the bottom row. The MSLP centre is also deeper for the Northern Hemisphere composites, but not for the Southern Hemisphere composite (consistent with Fig. 7(c,f) and 12(a)). The peak wind speed region is equatorward of the composite centre, extending from the warm sector rearwards behind the cold front towards the tip of the cloud head as well as along the cold front. A warm seclusion is evident even in the smooth SJP cyclone composite at this maximum intensity time in all the regions (although a closed $\theta_w$ contour is not evident for the North

Atlantic region with the contouring used).

The differences between the composite SJP and nSJP cyclones are strongest for the North Pacific cyclones in central MSLP and peak low-level wind speed. The peak differences are found on the poleward edge, i.e., closer to the cyclone centre, of the region of strongest winds in the SJP cyclones (as can be seen in bottom row of Fig. 8 by comparing the shaded wind differences





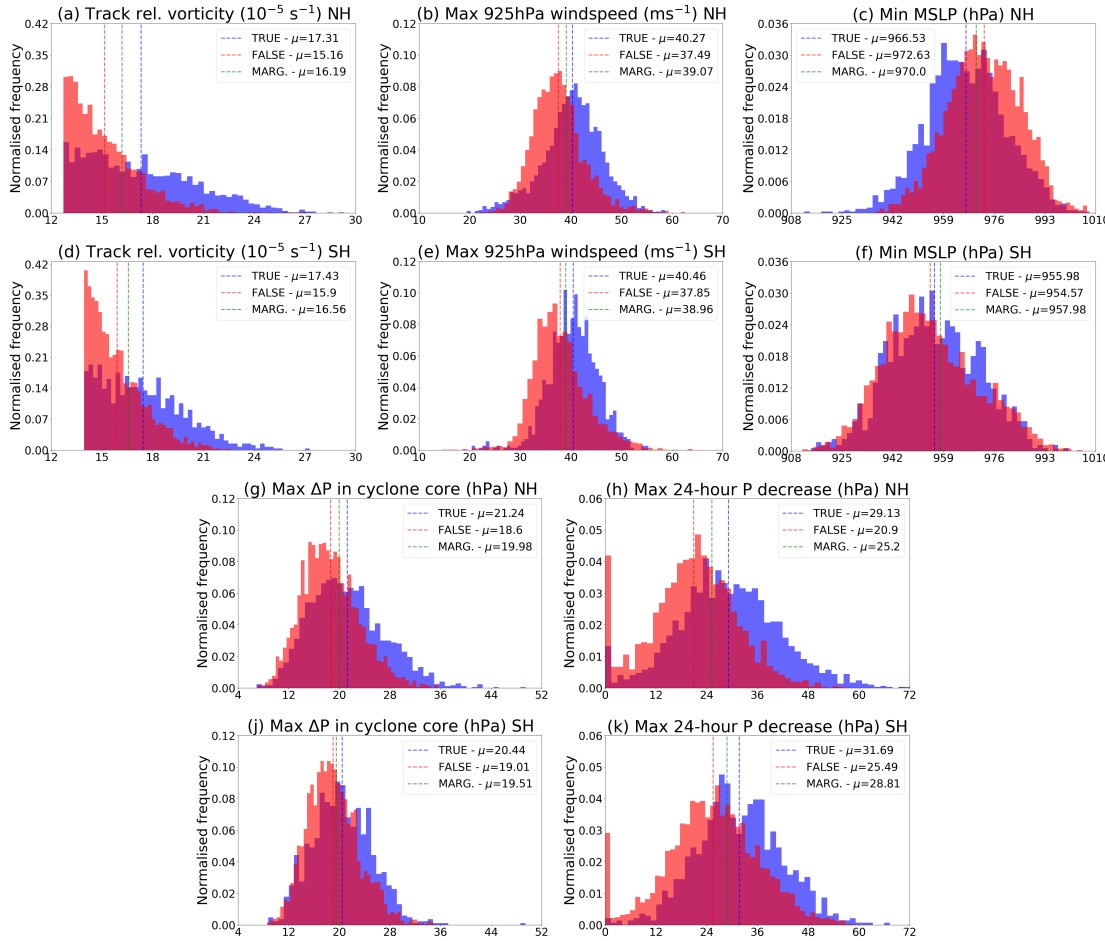

**Figure 7.** Normalised frequency plots of SJP and nSJP cyclone intensity metrics. Orange and purple bars are for SJP and nSJP cyclones, respectively, with the area covered by both sets of bars in dark red. Mean values of cyclone sets are indicated by dashed vertical lines, with the values also listed in the legend, where TRUE, FALSE and MARGINAL labels indicate SJP, nSJP and marginal cyclones, respectively. Panels (a,b,c,g,h) and (d,e,f,j,k) are for the Northern and Southern Hemispheres, respectively. Intensity metrics in (a-g,j) are considered at the time of maximum intensity of the cyclone (according to $\xi_{850}$). (a,d) Track maximum $\xi_{850}$. (b,e) Maximum wind speed at 925 hPa within a $6°$ geodesic great circle radius circle centred on the track (as a simple grid point search). (c,f) Minimum MSLP within a $5°$ geodesic great circle radius circle centred on the track using a B-spline interpolation and steepest descent minimisation. (g,j) Maximum difference in MSLP in a 300-km-radius circle centred at the location of minimum MSLP. (h,k) Maximum 24-hour decrease in MSLP along the full length of the track; this value is set to zero in absence of any 24-hour decrease.

with the stippled region indicating the location of the composite SJP cyclone peak winds) and elongates along the composite
cold front. Although the wind speed differences between the composite SJP and nSJP cyclones are overwhelmingly positive,
for the Southern Hemisphere cyclones there is a negative anomaly of wind speed poleward of the cyclone centre, where the cold




conveyor belt travels rearward relative to the cyclone motion, approaching the magnitude of the positive anomaly in the peak wind region. This negative anomaly is associated with a stronger MSLP pressure gradient in this region in the composite nSJP cyclone (as can be inferred from the MSLP difference field) and so consistent with stronger gradient-wind-balanced winds in the cold conveyor belt jet. Note though that this negative difference in wind speed in the Southern Hemisphere composite, and also to some extent the positive differences in the peak wind speed in all three regions, are also partially attributable to the slower cyclone travel speed of the composite nSJP cyclones, as discussed further below.


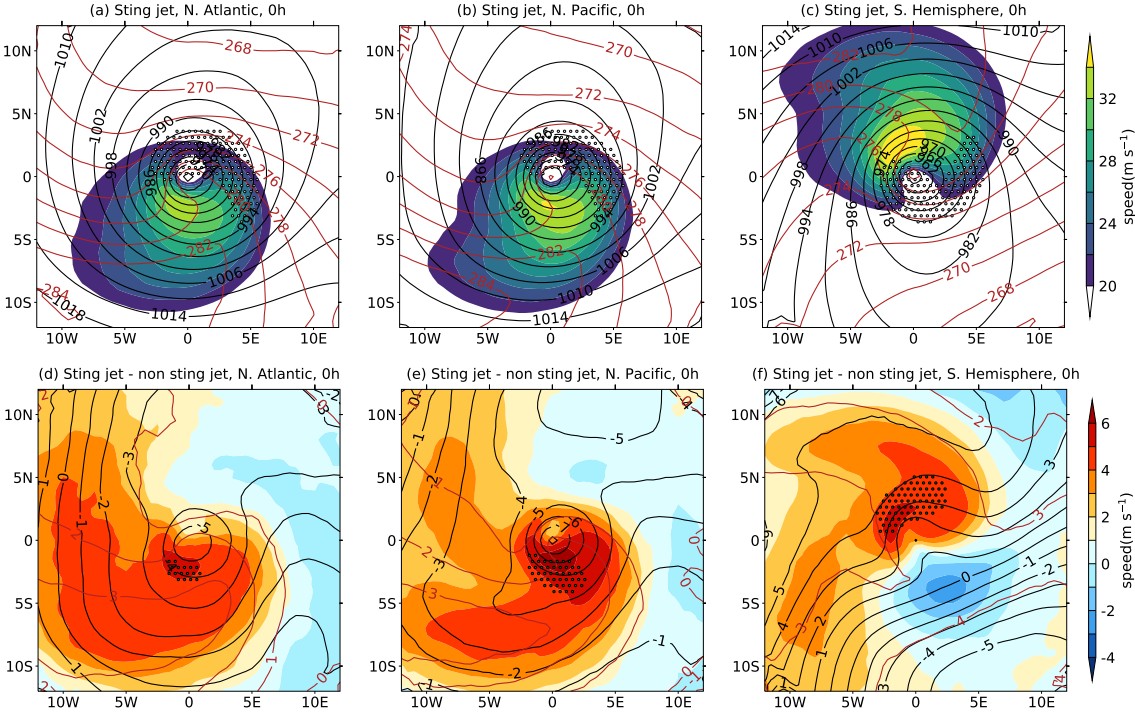

**Figure 8.** Maps of the low-level composite structure of the SJP cyclones (top row) and their differences to the nSJP cyclones (bottom row) at the time of maximum intensity for the (a,d) North Atlantic, (b,e) North Pacific, and (c,f) Southern Hemisphere: (top row) 850-hPa wind speed (shaded), MSLP (black contours), 850-hPa $\theta_w$ (brown contours) and 700-hPa cloud (stippling indicates relative humidity with respect to ice >80%; (bottom row) difference fields for wind speed (shaded), MSLP (black contours) and 850-hPa $\theta_w$ (brown contours) with green stippling marking the region of winds exceeding 32 m s$^{-1}$ in the equivalent SJP composites.

The evolution of the low-level North Atlantic composite cyclone structure over the 24 h prior to maximum intensity is shown in Fig. 9 and Fig. 8(a,d) for 24–6 h and 0 h prior to the time of maximum intensity, respectively. As expected, the centre of the composite SJP cyclone deepens in MSLP as the cyclone intensifies. The warm sector (demarcated by the $\theta_w$ contour of 284 K at earlier times, and 282 K at later times, in the top tow) narrows, mainly due to the cyclonic rotation of the cold front in these rotated composites, as the composite cyclone matures from an open wave towards a secluded cyclone. The core $\theta_w$ decreases as the cyclone matures, consistent with the poleward cyclone motion (see Fig. 12(d)). At the start of the time period shown




the composite cold front is noticeably stronger in terms of $\theta_w$ gradient than the composite warm front. However, the warm

front strengthens as the cyclone matures such that by maturity the strength of the warm front approaches that of the cold front. The magnitude of the winds in the strong-wind region increases and the region of peak winds moves from being located in the centre of the warm sector to straddling the cold front as the composite cyclone matures, consistent with the expected increasing dominance of the cold conveyor belt jet relative to the warm conveyor belt jet (see Fig. 1 of Hewson and Neu (2015)). The differences in central MSLP and peak wind speed between the composite SJP and nSJP cyclones increase as the cyclones

approach maturity. As well as the developing central MSLP pressure difference between the composite cyclones, the MSLP is far less in the SJP, compared to the nSJP, composite to the northwest of the cyclone centre, with this difference largest at the earliest time shown when it exceeds 6 hPa in the domain corner (the MSLP fields can also be compared using Fig. 10 which shows the fields for the composite nSJP cyclones in the bottom row). It is hypothesized that this difference results from a greater proportion of the SJP cyclones forming as secondary cyclones such that the lower pressure to their northeast is due to

the associated primary cyclone, but this hypothesis has not been tested here. Equivalent figures to Fig. 9 are shown in Figs. S1 and S2 for the North Pacific and Southern Hemisphere basins, respectively, and show similar evolution characteristics to the North Atlantic.

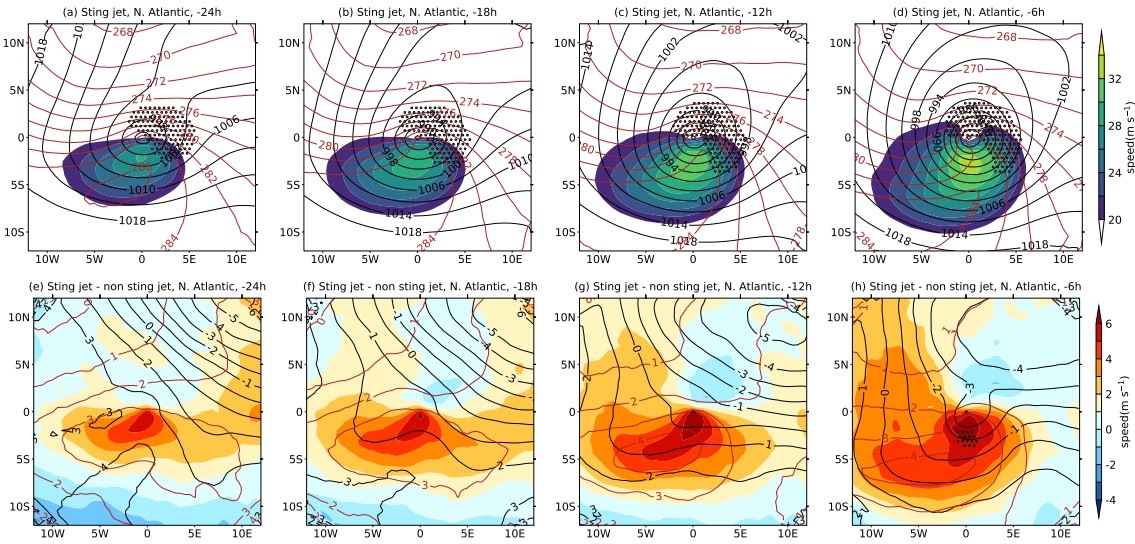

**Figure 9.** Maps of the low-level composite structure of the SJP cyclones (top row) and their differences to the nSJP cyclones (bottom row) for the North Atlantic at (a,e) 24, (b,f) 18, (c,g) 12, and (d,h) 6 h prior to the time of maximum intensity. Fields shown as in Fig. 8.

The evolution of the upper-tropospheric (300-hPa) wind structure for the composite North Atlantic SJP and nSJP cyclones is shown in Fig. 10. The upper-level jet is notably stronger in the composite SJP cyclone and while the cyclone centre moves

northwards relative to the jet core as the cyclone intensifies in both sets of composites, in the SJP composite it moves from being embedded in the jet core to the left jet exit region (a preferential region for cyclone intensification due to enhanced vortex stretching) whereas in the nSJP composite it is in the jet exit region throughout. This difference suggests that cyclones with the



SJP are more likely to have crossed from the equatorward to poleward side of the jet as they intensify, and so be located in the right jet entrance region earlier in their development (another preferential region for cyclone intensification due to enhanced vortex stretching). Equivalent figures to Fig. 10 are shown in Figs. S3 and S4 for the North Pacific and Southern Hemisphere basins, respectively. The composite upper-level jet is strongest overall for SJP cyclones in the North Pacific and the cyclone centre also begins more equatorward relative to the jet core in the SJP, compared to the nSJP, composites in these other basins.

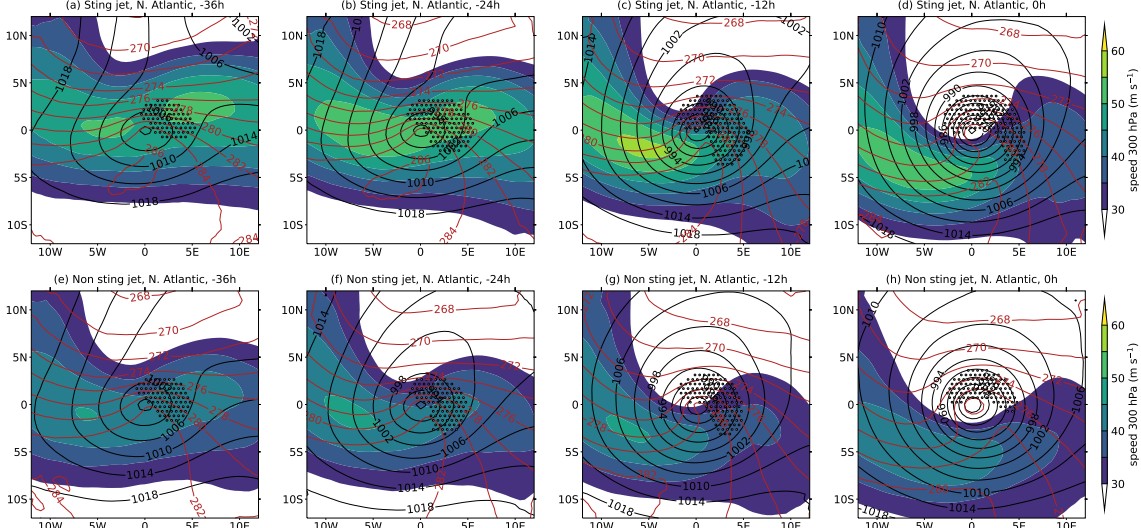

**Figure 10.** Maps of the composite upper-tropospheric wind structure of the SJP (top row) and nSJP (bottom row) cyclones for the North Atlantic at (a,d) 24, (b,e) 12 and (c,f) 0 h prior to the time of maximum intensity. Fields shown as in the top row of Fig. 9 but instead with the 300-hPa wind field shaded.

The mid-level PV structures of the composite North Atlantic cyclones are compared in Fig. 11. A strong PV anomaly exists in the centre of the composites at the time of maximum intensity, extending radially towards the cloud edge to the north and east, with PV values exceeding 1 PVU (Figs. 11(d,h)). This localised anomaly is embedded in a general gradient of PV across the domain from lower values in the south to higher values in the north (with extreme values towards the southwest and northeast corners of the domain). This gradient is consistent with the expected background PV gradient due to the latitudinal dependence of the Coriolis parameter; the rotation of this background gradient from south to north is due to the rotation of the cyclones prior to compositing as this rotation is generally slightly clockwise due to the typical northwestward cyclone motion. The anomalously large mid-level PV at the composite centres is consistent with strong production from diabatic cloud processes, a known feature of intense extratropical cyclones (e.g., Čampa and Wernli, 2012). The PV has larger values in the SJP composite with the maximum central value (within 300 km of the composite centre) exceeding that in the nSJP composite by more than 0.2 PVU at this time (Fig. 12(c)), implying stronger generation from diabatic processes. There is also a difference in the background PV between the two composites, with values in the SJP composite being generally lower across the domain;



the magnitude of this difference exceeds 0.5 PVU in the north of the domain. This difference arises from the generally more
more equatorward location (associated with smaller Coriolis parameter values) of the SJP cyclones.

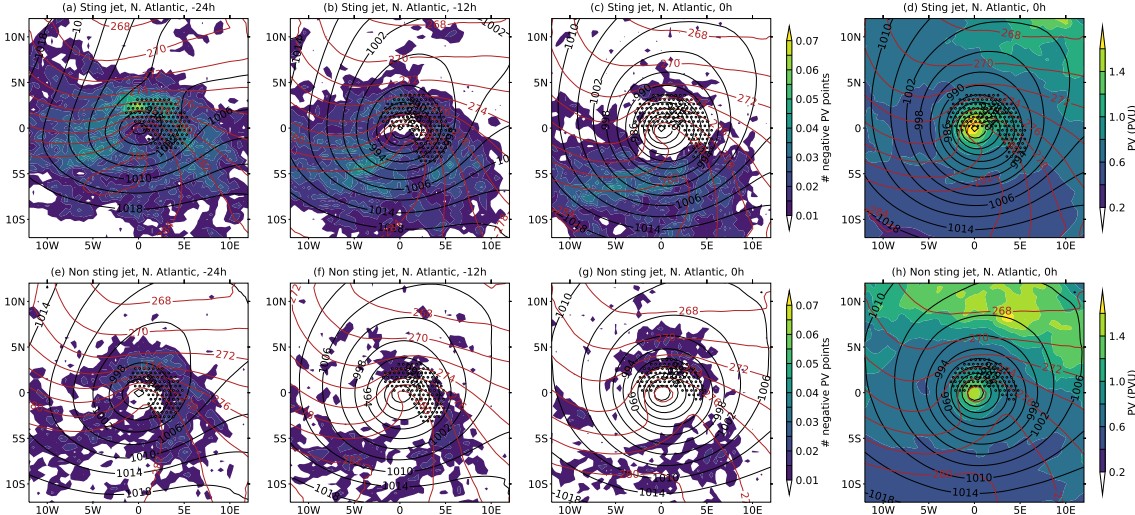

**Figure 11.** Maps of the composite 700-hPa PV structure of the SJP (top row) and nSJP (bottom row) cyclones for the North Atlantic. In (a–c,e–g) shading is the composite number of negative (symmetrically unstable) PV points at (a,e) 24, (b,f) 12 and (c,g) 0 h prior to the time of maximum intensity; in (d,h) shading is the composite PV at the time of maximum intensity. All other fields as in the top row of Fig. 9.

The rest of the panels in Figs. 11 show the evolution of a PV measure of mesoscale instability. Although the composite PV is positive throughout the domain for both sets of composites, individual cyclones have localised regions of negative PV. Negative PV implies the existence of (dry) symmetric instability, provided the environment is inertially and gravitationally stable, and

has been previously found in individual case studies of cyclones containing sting jets (Volonté et al., 2018, 2020, 2023b), often forming in narrow bands within the cloud head elongated along the direction of the bent-back front and tracking towards the cloud head tip as the cyclone develops. Hence the existence of negative PV is an alternative, independent diagnostic for mesoscale instability to the CSI-based diagnostic that has been used to classify the cyclones according to the likely existence of a sting jet within them. The narrow and transient nature, and typically small magnitude compared to nearby regions of

positive PV, of the negative PV regions means that they are not visible in the composite PV fields. Instead, here the panels show the average number of points with negative PV in the composites. For example, peak values of this metric for the North Atlantic composites (which occur in Fig. 11(a)) approach 0.07 (i.e., 7%). In the SJP composites two regions with larger values of negative PV points are visible (Fig. 11(a–c)). The peak values are at the tip of the cloud head at 24 h prior to the time of maximum intensity and are within a band of large values that wraps around the north of the cyclone centre aligning with the

outer edge of the cloud head. The second region of larger negative PV points lies in a band along the cold front (along the 284 K $\theta_w$ contour). There is also a local enhancement of the number of negative PV points in the region between the cloud head tip and cold front. While bands of enhanced negative PV points continue to exist aligned along the cold front at 12 and 0 h prior to the time of maximum intensity in the SJP composite, the band aligned with the cloud head decays markedly over



this time period consistent with a stronger rate of release than generation of the mesoscale instability giving rise to negative
PV in this region. There are far fewer negative PV points in the cloud head and cold front regions in the nSJP of composites. Although there is some local enhancement the nSJP composites, with values peaking within the cloud head directly north of the composite centre at 24 h prior to maximum intensity, it is clear that there is substantially more mesoscale instability in the SJP composites as is expected given the mesoscale-instability nature of the precursor diagnostic. The same composite symmetrically unstable PV features and feature evolution are present to some degree in the other two basins (Fig. S5) although,
relative to the North Atlantic, there are substantially more and fewer points in the SJP composites for the North Pacific and Southern Hemisphere regions, respectively. In particular, bands aligned with the cloud head and cold front exist that decay and do not decay with time, respectively, in all three basins.

Finally, time series of composite field statistics are shown in Fig. 12 for times from 36 to 0 h prior to the time of maximum composite intensity and all four regions: North Atlantic, North Pacific and Northern and Southern Hemisphere. These time
series enable more quantitative analysis of some previously identified characteristics (i.e., minimum MSLP evolution) and evaluation of some new diagnostics. The composite SJP cyclones deepen more rapidly than the composite nSJP cyclones in all regions from 24 h prior to maximum intensity (Fig. 12(a)), although, consistent with Figs. 7(f) and 8(f), in the Southern Hemisphere the minimum MSLP pressure is similar in both composites at the time of maximum intensity as a consequence of the differences in composite MSLP at the start of the rapid deepening period. The differences in the travel direction and
speed between the SJP and nSJP composites are also consistent for the four regions (Fig. 12(d)). The composite SJP cyclones travel faster throughout the 36 hours considered and change from initially travelling in a more zonal direction, compared to the composite nSJP cyclones, to travelling in a more poleward direction as they approach their maximum intensity.

Composite SJP cyclones have a warmer core with the maximum 850-hPa $\theta_w$ (within 300-km of the composite cyclone centre) being greater than 1.7 K (and up to 3.9 K) warmer in all regions throughout the time period shown (Fig. 12(b)). In all four
regions there is a much larger increase in cloudiness (measured here as the number of 700-hPa cloudy points within 600 km of the composite cyclone centre) as the composite cyclones approach maturity in those with the SJP (Fig. 12(e)) although, due to differences in cloudiness at the start of the time period shown, only in the Northern Hemisphere regions is the cloudiness greater in the SJP composites at the time of maximum maturity. The warmer core at the start of time period shown is consistent with the genesis regions of the SJP cyclones being displaced towards the equator relative to those of the nSJP cyclones (Fig. 5(b,d)).
At maximum intensity time the SJP cyclones are found poleward of the nSJP cyclones on average (not shown) and so the warmer core is then consistent with a stronger warm seclusions in the SJP cyclones. The development of increased cloudiness in the composite SJP cyclones, relative to the composite nSJP cyclones in the Northern Hemisphere regions is not surprising given that the diagnostic used to classify the precursor cyclones is based on the existence of a mesoscale instability that occurs in cloud.

Next PV metrics are considered. The enhanced mid-level PV in the composite SJP cyclone core, relative to that in the nSJP composite, shown for the North Atlantic at maximum intensity time in Fig. 11(d,h), develops during the rapid deepening phase in all four regions (Fig. 12(c), noting that of course the minimum rather than maximum PV is considered for the Southern Hemisphere). In contrast, the enhanced numbers of symmetrically unstable PV points for the composite SJP cyclones are



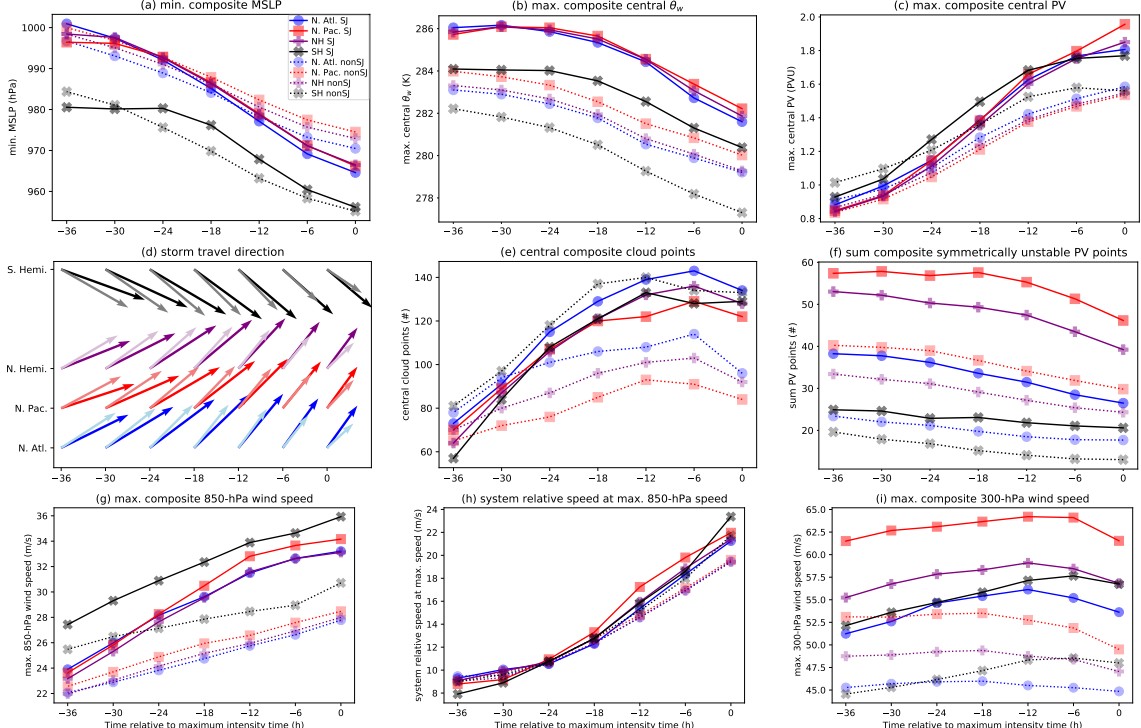

**Figure 12.** Time series of composite field statistics: (a) minimum MSLP, (b) maximum $\theta_w$ within 300 km of the composite MSLP centre, (c) maximum 700-hPa PV within 300 km of the composite MSLP centre (minimum value for Southern Hemisphere), (d) storm travel direction, (e) number of cloudy points (700-hPa relative humidity with respect to ice >80%) within 600 km of the composite MSLP centre, (f) total number of composite mean points with symmetrically unstable 700-hPa PV (negative and positive PV for Northern and Southern Hemisphere, respectively), (g) maximum 850-hPa wind speed, (h) 850-hPa system-relative wind speed at the location of the maximum 850-hPa (Earth-relative) wind speed, and maximum 300-hPa wind speed. In (d) the dark and pale arrows are for the sting-jet and non-sting-jet cyclones, respectively; in all other panels the solid and dotted lines are for the sting-jet and non-sting-jet cyclones, respectively.

present throughout the time period shown with the number of points slowly decreasing with time for most composites, with the

possible exception of the Southern Hemisphere SJP composite, (Fig. 12(c)). This decrease is seen despite that all points within the composite are counted in Fig. 12(c), but only at the cloud head tip was a strong decay seen in Figs. 11 and S5.

The bottom row of Fig. 12 shows wind speed time series. The maximum 850-hPa wind speed is clearly faster for the SJP composites with the differences between the two composites for the same region increasing during the 36-h intensification period considered (Fig. 12(g), consistent with the maps in Figs. 9, 8, S1 and S2). However, as shown in Fig. 12(i) (and

consistent with Figs. 10, S3 and S4), the upper-tropospheric jet is also substantially faster in the precursor composite leading to the faster travel of the SJP cyclones evident in Fig. 12(d). To address the question of whether the 850-hPa winds are stronger in the SJP cyclones simply because they are travelling faster, Fig. 12(h) shows the 850-hPa system-relative wind speed at the same location at the 850-hPa (Earth-relative) wind speed. While the system-relative wind speed increases in all the composites





as the cyclones intensify, the speeds for the composites with the SJP diverge from (becoming faster than) those for the nSJP
composites for each region during the last 18 h of intensification.

## 3.6    The representation of near-surface winds in SJP and nSJP cyclones from CCMP3.0 blended ocean winds

The differences between the SJP and nSJP cyclones have been presented in Sects. 3(3.2)-(3.5) using the ERA5 reanalysis
dataset. Here comparison is instead made using observations. As the vast majority of cyclones occur over the oceans, compar-
ison of SJP and nSJP cyclones using direct observations of near-surface wind speeds is hindered by a lack of high-resolution,
high-frequency global observations. The CCMP3.0 dataset provides the best available data, by blending bias-corrected ERA5
near-surface winds with scatterometer observations. To avoid including cyclones with limited or no observational input, we
identify the location of the maximum CCMP3.0 wind speed within a 10° box of each cyclone centre at the time of maximum
intensity, and filter to just those cyclones with at least two satellite observations contributing to the blended product. Figure 13
shows that the average maximum observed wind speed at the time of maximum intensity increases from nSJP cyclones to
marginal cyclones to SJP cyclones. This increase is consistent with Figs. 7(b,e) and 12(g) (for 925-hPa and 850-Pa wind speed,
respectively), confirming that SJP cyclones tend to have stronger wind speeds than nSJP cyclones. Furthermore, it is found that
the CCMP3.0 minus ERA5 difference in maximum near surface wind speed is larger also for the SJP cyclones than the other
two categories (not shown). This finding may be evidence that there is stronger mesoscale wind activity in the SJP cyclones,
e.g., associated with sting jets, which is observed in CCMP3.0 but not resolved by ERA5. However, given the non-linear nature
of the bias correction applied to the ERA5 winds to produce the CCMP3.0 background field, confirming this result requires
further work.

## 4    Conclusions

The aim of this study was to produce the first climatology of extratropical cyclones containing the transient mesoscale descend-
ing jet termed a sting jet and so explore their existence beyond the North Atlantic-European region, where the vast majority of
previous research (including climatologies and case studies) has been focused. An algorithm for precursors to sting jets (SJP)
has been refined using a set of expertly assessed notable storms and applied to more than 10,000 tracked cyclones identified in
43 years of ERA5 data. This cyclone dataset consisted of those containing a warm seclusion, as diagnosed by a new method
employing a watershed algorithm, in the top intensity decile of all tracked ERA5 cyclones. The key findings of this study
address the two questions posed in Sect. 1 as follows:

1. Cyclones containing sting jets are likely to be present in all major ocean basins, despite a lack of published case analyses
       outside of the North Atlantic-European region.

       – Globally the percentages of the diagnosed warm-seclusion cyclones in the top intensity decile with the SJP, without
          the SJP and marginal cases are 29%, 55%, and 16%, respectively. If all of the cyclones without a warm-seclusion
          in this top intensity decile are assumed to not yield a sting jet (because of the absence of a weak stability frontal-



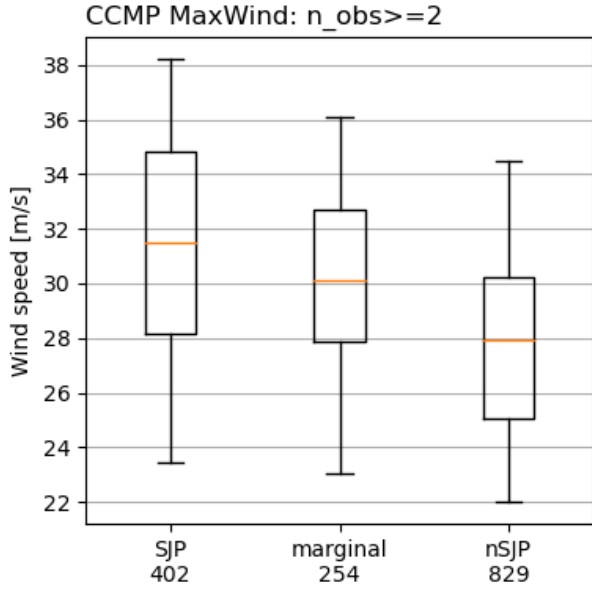

**Figure 13.** Maximum CCMP3.0 wind speeds within a 10° box of each cyclone centre at the time of maximum intensity. All cyclones from the CCMP3.0 period (1993–2019) that have at least two independent scatterometer observations contributing to the blended wind speed product at the grid point of the maximum are included.

fracture region) then 21% of the top intensity decile cyclones may have a sting jet. However, SJP are far more common in the Northern Hemisphere (37% of warm-seclusion cyclones, 27% of all cyclones) than in the Southern Hemisphere (20%, 15%), with similar values for the North Atlantic and North Pacific basins as for the whole Northern Hemisphere. The precise percentages will be dependent on the thresholds chosen for the SJP diagnostic (albeit that they have been refined using a holistically assessed set of notable storms). Analysis of all cyclones (all

intensities and both with and without a diagnosed warm seclusion) for a single extended-winter season over the North Atlantic has revealed that the precursor is equally likely to be present in cyclones with and without a warm seclusion. As the sting jet is unlikely to be able to descend towards the surface in cyclones that do not follow the Shapiro-Keyser evolution with the associated frontal fracture and warm seclusion, this result implies that estimates of sting jet prevalence based on instability metrics only (such as in Hart et al. (2017)) are likely overestimates.

– SJP cyclones are more likely to have their genesis nearer the equator than nSJP cyclones (in both hemispheres) and form over the warm western boundary currents in the Northern Hemisphere. Composite SJP cyclones track faster than nSJP cyclones and initially track more zonally, before changing to have a greater poleward component, than nSJP cyclones in all regions. In the North Atlantic, the difference in track density between SJP and nSJP cyclones found here is consistent with the previous climatology for this region (Hart et al., 2017).





– In deriving the climatology it was also found that the proportion of cyclones with a warm seclusion increases strongly with decile of maximum intensity (and is similar for all regions) such that about 75% of cyclones in the top intensity decile have warm seclusions. This result implies a strong relationship between cyclone structure and intensity.

2. Cyclones with and without the SJP have distinct characteristics in terms of their spatial and temporal variability, distri-
butions of various intensity metrics, and composite structure. Here we reiterate that the ERA5 reanalysis data used is produced using a model that has insufficient resolution to represent sting jets and the vast majority of these cyclones form over the oceans where observational data, especially at low levels, is very limited. Also, the surface footprint of sting jets, estimated as perhaps 50–100 km across, would barely be represented in this dataset with ∼30 km grid spacing even if sting jets were represented. Hence, the differences found between SJP and nSJP cyclones in the ERA5 dataset
are not due to sting jet presence *per se*, but instead reveal larger-scale differences between cyclones with and without the SJP. While specifically the SJP is a diagnosis of substantial CSI in the cyclone cloud head, more generally it likely distinguishes those cyclones with strong cloud-related diabatic processes.

     – SJP cyclones are globally more intense than nSJP cyclones in terms of maximum $\xi_{850}$, maximum low-level wind speed, and maximum MSLP anomaly at the time of maximum intensity (in terms of $\xi_{850}$) as well as in terms
of maximum 24-h MSLP decrease. Whereas Northern Hemisphere SJP cyclones are on average 6 hPa deeper in MSLP at the time of maximum intensity, in the Southern Hemisphere the values are similar. The latter, apparently anomalous, result is a consequence of different tracks typically taken by Southern Hemisphere SJP and nSJP cyclones.

     – Comparison of the composite structure of SJP and nSJP cyclones reveals significant structural differences. In addi-
tion to the differences highlighted above, particularly notable are that the composite SJP cyclones have a warmer (higher values of 850-hPa $\theta_w$) core than the nSJP cyclones in all regions and that the composite MSLP field is consistent with a greater proportion of SJP cyclones forming as secondary cyclones. The composite upper-tropospheric jet (at 300 hPa) is substantially faster for the composite SJP cyclones throughout the 36 h prior to maximum intensity time considered (typically by about $10\,\mathrm{m\,s^{-1}}$) and SJP cyclones are more likely to cross from the equatorward
to poleward side of the jet as they intensify. Analysis of the system-relative wind speeds shows that the faster (Earth-relative) low-level wind speeds found in the SJP cyclones are greater than expected due solely to the faster cyclone travel speed.

     – Analysis of the composite PV structure of the cyclones reveals larger maximum (minimum in Southern Hemisphere) mid-level PV values in the SJP cyclone cores, consistent with stronger PV generation through cloud latent
heating. There are also more points with symmetrically unstable PV (negative in the Northern Hemisphere and positive in the Southern Hemisphere) in the SJP cyclones, providing further evidence of enhanced mesoscale instability in these cyclones. The number of points in the cloud head tip decreases with time, implying that the instability is released.



– Finally, comparison of cyclone maximum wind speeds derived using a dataset consisting of a blend of scatterometer
and bias-corrected ERA5 data for available cyclones show that the median wind speed for the SJP cyclones is
more than $3\,\mathrm{m\,s^{-1}}$ greater than that for the nSJP cyclones, providing independent verification of the wind speed
differences between the two cyclone categories.

The implications of these findings are that sting jet cyclones are ubiquitous in the extratropics (in both hemispheres) despite
published detailed case studies of these cyclones to date only existing for cyclones in the North Atlantic-European regions. The
potential enhancement of near-surface winds and gusts due to sting jets, which may be poorly represented in weather forecast
models, has implications for weather warnings due to land-falling cyclones. The greater intensity of cyclones containing the
SJP diagnostic in ERA5 data (compared to those without the SJP) and structural differences revealed by the compositing reflect
the important contribution of cloud diabatic processes (leading to CSI and other mesoscale instabilities) in cyclone development
with implications for changes in the intensity of cyclones in a warmer future climate where cloud diabatic processes are likely
to be enhanced.

Open questions remaining include (i) why SJP cyclones are far less prevalent in the Southern Hemisphere than in the
Northern Hemisphere and the consequence of this difference for the presence of sting jets, (ii) whether sting-jet cyclones
occur beyond the extratropical band considered here, (iii) the relationship between the presence (and magnitude) of the SJP,
the presence of sting jet(s), and enhanced surface winds due to sting jet(s) in cyclones, and (iv) whether, despite a lack of
published analysed case studies, sting jets could occur in cyclones that meet the criteria for the SJP and yet follow an evolution
far removed from the classical Shapiro-Keyser evolution with a warm seclusion and frontal fracture. In particular, the latitudinal
constraints used for the cyclone tracks (40–80°) exclude much of the Mediterranean Sea where diabatic processes are known
to be particularly important for the development of some cyclones (Flaounas et al., 2021) and where strongly convective
medicanes with tropical characteristics occur. The difference in characteristics of the cyclones in this region compared to those
in the main storm tracks, including typically smaller sizes and shorter lifetimes, mean that further work would be needed to
refine and verify the SJP for this region. It is anticipated that the databases generated here of cyclones with and without SJPs
will enable case study and other analyses to be performed to address some of these questions. Finally, while outside the scope
of this study, interesting questions linked to conceptual models of cyclone development can be addressed using the generated
database of intense warm-seclusion cyclones.

*Code and data availability.* For the ERA5 data, Hersbach et al. (2023a) was downloaded from the Copernicus Climate Change Service
(2023a) and Hersbach et al. (2023b) was downloaded from the Copernicus Climate Change Service (2023b). The results contain modified
Copernicus Climate Change Service information 2020. Neither the European Commission nor ECMWF is responsible for any use that
may be made of the Copernicus information or data it contains. The warm seclusion algorithm was developed using the watershed image
segmentation algorithm provided by the scikit-image python library (van der Walt et al., 2014). The TRACK algorithm is available on the
University of Reading's Git repository (GitLab) at https://gitlab.act.reading.ac.uk/trachttps://gitlab.act.reading.ac.uk/track/k/track (Hodges,
2023). The satellite images used in Fig. 2 and Sect. 2.5 were obtained from the EUMETSAT Data Portal using the eumdac python library,





and the CCMP3.0 data was downloaded from Mears et al. (2022b). Code to calculate DSCAPE from vertical profiles is available from github at https://github.com/omartineza/csisounding.

*Author contributions.* SLG, OM-A and AV conceptualized and acquired funding for the study. All authors performed parts of the software
development and data analysis. SLG led the paper writing, but all authors drafted parts of the text and contributed to the editing and review process. BJH led the data curation.

*Competing interests.* The authors declare that they have no conflict of interest.

*Acknowledgements.* This work was funded by the UKRI Natural Environment Research Council (NERC), grant number NE/X010473/1. OM-A and BH also acknowledge funding from the NERC Climate change in the Arctic – North Atlantic region and impacts on the UK (CA-
NARI) programme, grant number NE/W004984/1. The authors thank Kevin Hodges for supplying the identification and tracks of cyclones from the ERA5 dataset as well as code to produce the cyclone density maps. We also thank Helen Dacre for her useful comments on a draft of this paper.



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
