# Peer review of "A global climatology of sting-jet extratropical cyclones"

_EGUsphere, 2024_

## Author Comment (AC1)

**Final author comments on the reviewers' comments on our paper "A global climatology of sting-jet extratropical cyclones" by Gray, Volonté, Martínez-Alvarado and Harvey**

In our response, the reviewer comments are in black font with our responses in blue.

**Reviewer 1 (Emmanouil Flaounas):**

I read the paper with great interest and I found it helpful and insightful, providing a richness of interesting results. A global climatology of sting-jet cyclones was indeed missing from the state of the art and this paper provides an overview of these systems' structural characteristics and insights into their dynamics. I have comments and suggestions of minor nature on the organization of the manuscript that I leave to the authors discretion if they want to follow them.

Many thanks, Manos, for your many positive comments. We're pleased that you liked our paper.

Section 2 is clearly described, albeit complicated and long. Complexity is not due to language or presentation, but due to the large amount of knowledge and techniques that someone needs to understand and be already familiar with. Probably sections 2.1, 2.3 and 2.6 could go together. Then, sections 2.2 and 2.4 could form a separate (sub)section which could be supported with schematics or flowcharts so that the reader can conceptually understand the role of different dynamical features in sting-jets. A subsequent subsection could use real cases as examples (now figures 1 to 3) to demonstrate the complexity of the exercise and what is expected in realistic datasets. Finally, section 2.5 although it is important it could be moved to supplement or an appendix.

Thank you for your suggestions. To make the methods section more easily "digestible", we have followed your advice in terms of combining the different parts of the methods section. We have split the methods section into two methods sections. The first methods section covers the reanalysis and observational datasets used, and the tracking and compositing methodology (previously sections 2.1, 2.3 and 2.6). The second methods section covers the warm-seclusion and sting-jet precursor algorithms (previously sections 2.2 and 2.4). The main part of the section on the assessment of the notable storms (previously section 2.5) has been moved to supplementary material (which is where the tables giving details of these notable storms can also be found). Figures 1-3 are now all contained within the second methods section. We have not added additional schematics or flowcharts showing the dynamical features in sting jets as such schematics can be readily found in other papers, in particular the review article of Clark and Gray (2018), which are referenced in our paper.

I am not sure there are many(any?) climatologies of warm seclusions (SJP+nSJP). It would be useful to know a bit more on how they compare spatially to other intense cyclones.

We very much agree and we had already noted in the final line of our conclusions section that "Finally, while outside the scope of this study, interesting questions linked to conceptual models of cyclone development can be addressed using the generated database of intense warm-seclusion cyclones.". We would prefer to leave this analysis to another paper rather than including a partial analysis here (a full analysis would overly extend this already substantial paper).

Section 3.1: If most SJP cyclones include a sting jet and if ERA5 may partly resolve these sting jets, then is it possible that sting jets are responsible for higher relative vorticity at 850 hPa in the center of the cyclones in T63 resolution? If so, then SJP cyclones should anyway be more intense than other cyclones (if we use relative vorticity as an intensity metric). A reply comes later in section 3.4, so I am wondering if this section should be merged with 3.4? (Please also see next comment on section 3.6).

The 31 km (TL639) resolution model used for ERA5 is likely too coarse to resolve sting jets according to the review of Clark and Gray (2018). However, the data assimilation step of producing the analyses ingests observations including scatterometer data and hence it is possible that there may be partial representation of sting jets in the reanalyses. In the paper text we have modified the statement that ERA5 is too coarse to resolve sting jets (see particularly point 2 of the conclusions). However, we do not think that any partial representation of sting jets in ERA5 will directly lead to the cyclones with the sting-jet precursors having higher relative vorticity than those cyclones without the precursor. The reasons for this are (i) the vorticity plotted in the histograms in Fig. 8a,d (previously Fig. 7a.d) are values taken from the filtered vorticity fields used for cyclone tracking (this has now been made clearer in the caption of the intensity metrics histograms figure) and so will not be strongly affected by a small-scale region of enhanced sting jet winds, (ii) these plots also use data from the maximum intensity (in terms of vorticity) time which is typically slightly later than when sting jets appear in low-level winds, (iii) cyclones with the sting-jet precursor are also diagnosed as more intense on average in terms of the depth of the mean-sea-level core (Fig. 8g), and (iv) cyclones with a sting-jet precursor were also found to be more intense in terms of relative vorticity in the North Atlantic climatology of Hart et al. (2017, see Fig. 3d) which used the coarser resolution ERA-Interim reanalysis dataset (model resolution TL255 equivalent to 80 km). Please see comment below regarding suggested merging of subsections.

Section 3.6: seems a bit weaker and detached from the rest of the paper and is only confirming that SJP cyclones include higher wind speeds. I am wondering if there should be a unique subsection that discusses the intensity of the cyclones involved in this analysis (i.e. merging current sections 3.1, 3.4 and 3.6).

Thank you for this suggestion. We have moved the material on observational analysis that was in Sect. 3.6 and combined it with that which was previously in Sect. 3.4 to create a section now called "Distributions of intensity metrics of SJP and nSJP cyclones in ERA5 and observations". We did not include Sect. 3.1 in this merger because Sect. 3.1 presents the characteristics of the full set of warm-seclusion cyclones and does not consider the split into those with and without the sting-jet precursor.

**Reviewer 2 (anonymous):**

This is a really great paper detailing a new climatology of sting jet cyclones considering all ocean basins (rather than just the North Atlantic region that has already received quite a bit of attention). The climatology is produced using a subset of intense cyclones that contain a warm seclusion. The sting jet precursor is calculated for each storm and some additional criteria used to determine the cyclones with the potential for producing sting jets. The analysis includes the spatial and temporal variability of the storms, the distributions of their characteristics, and composites of their structure.

The paper is well-written and contains thorough analysis and interesting results and conclusions. A key point is that sting jet cyclones seem to be possible in all ocean basins, and not just in the North Atlantic European region on which case studies have usually focused.

We thank the reviewer for their many positive comments about our paper.

I have a few minor comments and suggestions.

1. Line 43: Add reference for Norwegian model.

Reference to Bjerknes (1919) added.

2. Line 53: "There is published..." -> "there is no published...".

Corrected, thanks.

3. Lines 132-139: I think the watershed algorithm could be explained or described in a bit more detail. How are the watersheds defined?

A few lines of explanation of our implementation of the watershed algorithm have been added to the third paragraph of section 3.1 as follows:

" The watershed algorithm works by considering the $\theta_w$ values within this warm sector as an inverted local topography (i.e., $\theta_w$ maxima are valleys). The algorithm then "floods basins" from markers defined as local maxima in $\theta_w$ (lowest points of valleys) until the basins attributed to different markers meet on watershed lines, so defining the watersheds."

4. Line 151: Something strange with the section numbering – think this should be 2.5 (rather than 22.5).

Thank you for pointing out this LaTex glitch – the cross-references have been corrected.

5. Figure 1: This figure is a bit too small to see the necessary details in order to understand the method. Is there a way to make this larger?

We have made the figure larger by using the same colourbar for all three panels and consequently only showing 1, rather than 3, colourbars.

6. Line 164: This sentence doesn't seem to read correctly.

Corrected, thanks (also removed a couple of words from earlier in the paragraph to avoid repetition).

7. Figure 2 caption end – similar problem with section numbering.

See response to comment 4.

8. Line 337: Here it is pointed out that the nSJP cyclones have higher cyclogenesis in the secondary cyclogenesis regions. Later on it is suggested that the SJP cyclones have higher proportion of secondary cyclogenesis, indicated by the MSLP composites. It would be good to test the hypothesis given later (line 419), as these statements seem to be a little contradictory.

We agree that these comments are contradictory. We have looked again at the track and genesis density plots in Fig. 5 and decided that the genesis "hot spots" found towards the eastern ends of the North Atlantic storm track are too far north for the genesis to be in the region

where secondary cyclogenesis occurs. Hot spots exist off the tip of Greenland and to the east of Greenland/north of Iceland, regions more typically associated with lee cyclogenesis. We have thus edited the text referring to Fig. 5 to reflect this. We are considering additional diagnostics to assess the potential role of secondary cyclogenesis for the SJP cyclones.

9.  Line 354: How is the significance tested here?

The significance test used is described in the caption of the figure. It is a two-sided Wald Test.

10. Line 428: I think this is a really interesting result and it is nice to see it highlighted in the conclusions.

Thanks.

11. Line 502: "faster" -> "higher".

We disagree. It is correct to say that the "speed is clearly faster". We don't understand why "higher" would be better here.

---

## Author Response (AR2)

*HEAD OF SCHOOL*
Professor Andrew Charlton-Perez

School of Mathematical, Physical and Computational Sciences
**University of Reading,**
**Department of Meteorology,**
Meteorology Building, Whiteknights Road,
Earley Gate,
Reading, RG6 6ET, UK

*phone* +44 (0)118 378 6791
*email* s.l.gray@reading.ac.uk

24 October 2024

Dear Editors,

We are very pleased to hear that our paper has been accepted for publication. We have added longitude / latitude labels to our figure 2 as you requested in the uploaded files. Thank you for your work during this review process.

Yours sincerely,

Prof. Suzanne Gray